# OPTION: Optimal Transport–Guided Flow Matching for Incomplete and Unaligned Multi-View Clustering

**Siyuan Zhou** [1 2]   **Zhibin Gu** [1 3]

## Abstract

Multi-view clustering effectively exploits rich information from multiple views, yet real-world applications are frequently challenged by missing views and cross-view sample misalignment, hindering cross-view modeling and resulting inferior clustering performance. To address these challenges, this paper presents a novel method, **OP**timal **T**ransport–Gu**I**ded fl**O** Matchi**N**g for incomplete and unaligned multi-view clustering (**OPTION**). Specifically, OPTION employs conditional flow matching to learn deterministic transport paths for missing-view imputation, enabling stable manifold-preserving recovery and more discriminative representations. To support alignment-free fusion, we introduce a Gromov-Wasserstein-inspired structural regularization that aligns intra-view geometric structures in the latent space without solving hard correspondences. Furthermore, an optional contrastive regularization is incorporated to enhance cross-view consistency specifically for aligned settings. Extensive experiments demonstrate that OPTION outperforms state-of-the-art methods across ideal, incomplete, and unaligned scenarios evaluated separately. Code: https://github.com/TimoZhou1024/OPTION.

## 1. Introduction

Multi-view clustering (MVC) aims to partition samples into distinct groups by integrating complementary information from multiple heterogeneous data sources, and has attracted significant attention in machine learning and data mining communities (Chao et al., 2021; Chowdhury et al., 2025; Liu et al., 2025). However, most existing MVC methods are designed for ideal multi-view data, which assumes that all views are complete (every sample has observations in all views) and aligned (samples across different views have known one-to-one correspondences) (Xu et al., 2022; Zhang et al., 2023; Ren et al., 2024). In real-world applications, such ideal assumptions are often violated. For instance, in medical diagnosis, a patient may have CT scans but lack MRI images due to cost or availability constraints, resulting in incomplete views (Lin et al., 2021). In social media analysis, user profiles across different platforms (e.g., Twitter and LinkedIn) often lack explicit linking, creating naturally unaligned multi-view scenarios (Xin et al., 2025). Consequently, the problem of clustering multi-view data under simultaneous incompleteness and misalignment remains largely unresolved, posing a fundamental challenge to existing multi-view learning paradigms.

To address these challenges, recent studies have increasingly focused on developing effective methods for mining non-ideal multi-view data, thereby enabling robust multi-view clustering. For the incomplete multi-view scenario (IMVC), contrastive learning-based methods (Lin et al., 2021; Yang et al., 2023a; Jin et al., 2023) dominate recent literature, which recover missing view representations by maximizing cross-view feature consistency through instance-level or prototype-level alignment. More recently, Zhang et al. (Zhang et al., 2025) introduced diffusion models to IMVC, leveraging conditional denoising to generate high-fidelity missing views while enhancing clustering compactness. For the unaligned multi-view scenario (UMVC), Huang et al. (Huang et al., 2020) pioneered the permutation learning paradigm, formulating cross-view alignment as learning a differentiable matching network. Subsequent works (Xin et al., 2025; Guo et al., 2024) further improve robustness by exploiting multi-level structural guidance or noise-tolerant contrastive mechanisms to handle noisy and unpaired correspondences.

Although these methods have achieved promising performance in incomplete or unaligned multi-view clustering, they still suffer from several critical limitations. First, most existing methods treat incomplete and unaligned scenar-

---

[1]College of Computer and Cyber Security, Hebei Normal University, Shijiazhuang, China [2]School of Data Science, Fudan University, Shanghai, China [3]Key Laboratory of Tibetan Information Processing, Ministry of Education, Qinghai Normal University, Xining, China. Correspondence to: Zhibin Gu <guzhibin@hebtu.edu.cn>.

*Proceedings of the 43$^{rd}$ International Conference on Machine Learning*, Seoul, South Korea. PMLR 306, 2026. Copyright 2026 by the author(s).

ios as separate problems, employing distinct pipelines for view imputation and correspondence alignment. This two-stage paradigm inevitably leads to error propagation between stages and suboptimal multi-view fusion. Second, many generative approaches, such as diffusion-based IMVC methods (Zhang et al., 2025), rely on iterative refinement to reconstruct missing views, limiting their efficiency and responsiveness in dynamic or large-scale multi-view scenarios. Third, most existing UMVC methods (Huang et al., 2020; Xin et al., 2025) require explicit correspondence supervision or annotated sample pairs, which is expensive or infeasible when cross-view correspondences are inherently unknown.

To address the aforementioned limitations, this paper proposes **OPTION** (**OP**timal **T**ransport–Gu**I**ded Fl**O**w Matchi**N**g), a unified framework that supports missing-view generation, structure-level alignment, and clustering within a single end-to-end pipeline. The framework offers three key innovations. First, generation, alignment, and clustering are integrated into a unified objective function, where the flow matching module, GW-inspired structural alignment, and soft clustering are jointly optimized end-to-end, eliminating inter-stage error propagation. Second, conditional flow matching (Lipman et al., 2022) is employed to learn deterministic transport paths for missing-view imputation, enabling stable manifold-preserving recovery through ODE-based generation with only 10 integration steps and substantially fewer sampling iterations than diffusion models. Third, a structural regularizer inspired by GW is introduced to align intra-view geometric structures in the latent space without hard correspondence assignment, enabling alignment-free multi-view fusion for unaligned scenarios. Furthermore, an optional contrastive regularization is incorporated to enhance cross-view consistency specifically for aligned settings. In summary, our contributions are as follows:

- We propose OPTION, a unified framework that supports both incomplete and unaligned multi-view clustering through flow matching, achieving deterministic inference with significantly lower latency than diffusion-based methods.

- We develop a Gromov-Wasserstein-inspired structural regularization that enforces cross-view geometric consistency without relying on hard correspondence assignment, enabling alignment-free multi-view learning.

- We introduce a modular architecture that adaptively activates contrastive learning for aligned data while relying on GW-based structural alignment for unaligned scenarios.

- Extensive experiments on benchmark datasets demon-

strate that OPTION achieves competitive or superior performance compared to state-of-the-art methods while offering an average $8.1\times$ measured inference speedup over the diffusion-based baseline.

## 2. Related Work

Existing multi-view clustering (MVC) methods for imperfect multi-view data can be broadly categorized into two groups: methods for handling missing views and methods for unaligned views.

**Incomplete Multi-View Clustering.** Early approaches relied on matrix factorization, sparse modeling, or graph learning to impute missing information implicitly (Liu et al., 2023; Yang et al., 2023a). With the advent of deep learning, sophisticated representation learning techniques have become mainstream. Methods like COMPLETER (Lin et al., 2021), DeepIMVC (Yang et al., 2023b), and information recovery-driven networks (Liu et al., 2024) utilize contrastive learning, prototype alignment, or reconstruction-driven representation recovery to handle missing views. To further address complex semantic consistency, recent works propose hierarchical alignment and topological guidance. For instance, HSACC (Ding et al., 2025) and DIMVC-HIA (Du et al., 2026) utilize hierarchical semantic imputation to reduce error propagation, while SCVT (Dong et al., 2025) constructs an optimal transport-based view topology graph to selectively guide missing sample completion. Beyond representation-based approaches, generative models—especially Diffusion Models (DMs) (Ho et al., 2020; Lim et al., 2023)—have been introduced to MVC for their superior generation quality (e.g., DCG (Zhang et al., 2025)). Despite their success, diffusion-based and complex topology methods still suffer from high inference latency due to iterative denoising or rely heavily on paired data, limiting their efficiency in real-time scenarios.

**Unaligned Multi-View Clustering.** Most MVC algorithms assume a strict one-to-one correspondence across views. When this assumption is violated (i.e., unaligned MVC), performance degrades significantly. Existing solutions usually formulate alignment as a permutation learning problem. PVC (Huang et al., 2020) and its successors employ differentiable matching networks to infer correspondences. Recent works like MRG-UMC (Xin et al., 2025) and CANDY (Guo et al., 2024) introduce multi-level guidance and robust contrastive learning to handle noisy or unpaired data. However, these methods often involve solving the assignment problem (e.g., via the Hungarian algorithm), which scales poorly ($O(N^3)$) with dataset size. To mitigate this computational burden, the latest studies explore alignment-free paradigms. For example, AF-UMC (Sun et al., 2025a) avoids direct sample matching by extracting consistent representations through a shared basis space, and CPMN (Wang et al., 2025)

aligns high-level cluster prototypes instead of instance-level features. Furthermore, they typically treat alignment and clustering as separate stages or alternating objectives, lacking a unified framework to handle structural misalignment implicitly.

## 3. Proposed Method

### 3.1. Problem Formulation and Framework Overview

Consider a multi-view dataset $\mathcal{X} = \{\mathbf{X}^{(v)}\}_{v=1}^V$ with $V$ views. A binary mask $\mathbf{M} \in \{0, 1\}^{N \times V}$ indicates view availability (incomplete MVC), and cross-view sample correspondences may be unknown (unaligned MVC). Our goal is to jointly: (1) impute missing views, (2) align cross-view geometric structures without hard correspondence assignment, and (3) discover cluster assignments—all within a unified end-to-end framework.

To achieve this, OPTION adopts a unified generation-alignment-clustering paradigm with three core components: latent embedding that projects heterogeneous views into a shared low-dimensional space via view-specific encoder-decoders; Gromov-Wasserstein-inspired alignment that enforces cross-view geometric consistency without hard correspondence assignment; and conditional flow matching that enables efficient missing-view imputation with only 10 ODE steps. Figure 1 illustrates the pipeline.

### 3.2. Latent Space Construction

To address the high dimensionality and heterogeneity of multi-view data, we project all views into a unified low-dimensional latent space. For each view $v$, a neural encoder $f_{\phi_v} : \mathbb{R}^{d_v} \to \mathbb{R}^{d_z}$ maps the observation $\mathbf{x}_i^{(v)}$ to a latent representation $\mathbf{z}_i^{(v)} = f_{\phi_v}(\mathbf{x}_i^{(v)})$. Symmetric decoders $g_{\psi_v} : \mathbb{R}^{d_z} \to \mathbb{R}^{d_v}$ reconstruct the original views, providing a reconstruction loss $\mathcal{L}_{rec} = \sum_v \|\mathbf{x}^{(v)} - g_{\psi_v}(\mathbf{z}^{(v)})\|^2$ that regularizes the latent space. This design reduces computational complexity from $O(d_v^2)$ to $O(d_z^2)$ for flow matching operations while enabling meaningful cross-view comparisons in the shared space $\mathbb{R}^{d_z}$.

### 3.3. Gromov-Wasserstein-Inspired Structural Alignment

A fundamental challenge in unaligned multi-view clustering is that the $i$-th sample in view $v$ may not correspond to the $i$-th sample in view $u$. As a result, conventional alignment strategies—such as contrastive learning or direct feature averaging—are largely inapplicable, since they rely on explicit sample correspondences. To address this, we employ a structural alignment mechanism inspired by the Gromov-Wasserstein (GW) distance. Unlike the standard Wasserstein distance, which evaluates point-to-point transport costs,

GW compares pairwise relational structures. This relational perspective motivates us to regularize view-specific latent spaces through their intra-view geometry rather than through a hard one-to-one matching step.

To capture the geometric structure of each view, we construct a pairwise similarity matrix. Specifically, for each view $v$, we compute an RBF (Radial Basis Function) kernel matrix $\mathbf{K}^{(v)} \in \mathbb{R}^{B \times B}$ within a mini-batch of size $B$, where $B$ denotes the mini-batch size:

$$K_{ij}^{(v)} = \exp\left(-\gamma \|\mathbf{z}_i^{(v)} - \mathbf{z}_j^{(v)}\|^2\right) \tag{1}$$

where $\gamma > 0$ is a bandwidth parameter. The entry $K_{ij}^{(v)}$ quantifies the similarity between the $i$-th and $j$-th samples within view $v$: values close to 1 indicate neighboring latent samples, while values close to 0 indicate distant samples. Thus, $\mathbf{K}^{(v)}$ encodes local manifold geometry rather than point-wise correspondences. To avoid manual tuning and keep kernel values informative across training phases, we set the bandwidth adaptively by the median heuristic, $\gamma = 1/(2\sigma_{med}^2 + \epsilon)$, where $\sigma_{med} = \mathrm{median}(\{\|\mathbf{z}_i - \mathbf{z}_j\|\}_{i<j})$ is computed within each mini-batch and $\epsilon$ is a small numerical constant. This kernel formulation is robust to feature scale variations across heterogeneous views.

The core insight is that if two views capture the same underlying cluster structure, their kernel matrices should exhibit similar patterns—even when sample indices are shuffled. We formalize this via the GW-inspired loss:

$$\mathcal{L}_{gw} = \frac{2}{V(V-1)} \sum_{v<u} \|\mathbf{K}^{(v)} - \mathbf{K}^{(u)}\|_F^2 \tag{2}$$

This batch-wise surrogate penalizes discrepancies between the pairwise relationship structures across all view pairs. Intuitively, minimizing $\mathcal{L}_{gw}$ encourages the encoder to learn representations such that the intra-view geometric structure becomes consistent across views, thereby providing implicit structural alignment without solving a hard correspondence problem. Importantly, this formulation avoids the $O(N^3)$ assignment-style cost of global matching and uses an $O(B^2)$ mini-batch regularizer. Its robustness comes from three design choices: mini-batch structural snapshots reduce the influence of large-scale noisy pairings, soft kernel similarities avoid brittle hard matching such as Hungarian assignment, and joint optimization with reconstruction, clustering, and flow losses prevents structural alignment from being dominated by spurious pairwise patterns. If view-specific neighborhood structures are severely corrupted, this term may still provide imperfect guidance; we therefore use $\mathcal{L}_{gw}$ as a scalable GW-inspired relaxation, not as a full discrete GW solver.

### 3.4. Conditional Flow Matching for Imputation

Once the latent space is established, missing view imputation becomes essential for downstream fusion and clus-

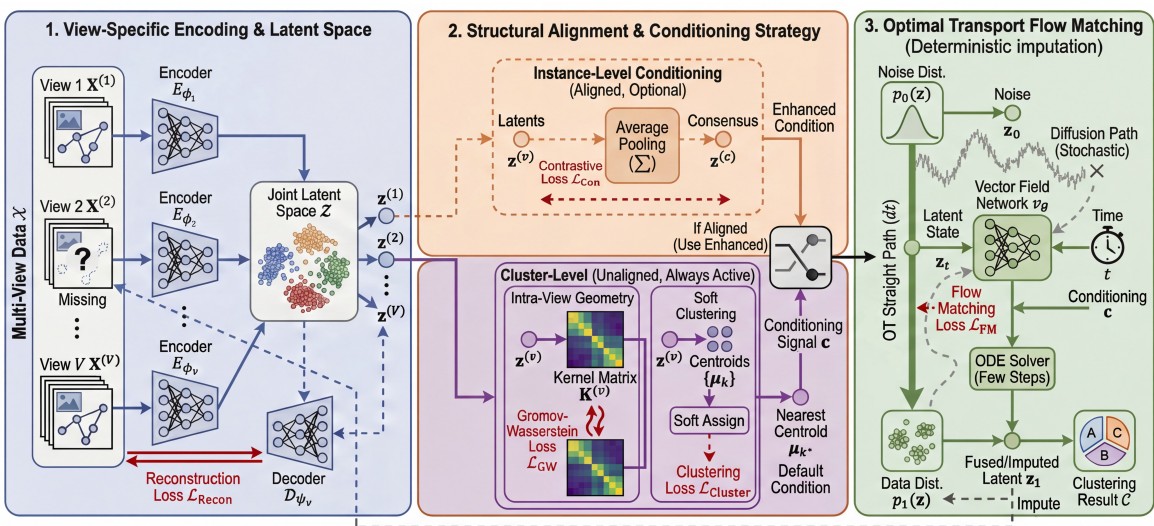

*Figure 1.* Overview of the proposed OPTION framework. The architecture consists of view-specific encoders that project heterogeneous inputs into a unified latent space, a conditional flow matching module that learns optimal transport paths from noise to data conditioned on semantic anchors, and a Gromov-Wasserstein-inspired structural alignment mechanism that enforces cross-view geometric consistency without hard correspondence assignment.

tering. While diffusion models have shown strong generative capabilities, their iterative denoising process (typically $T = 100 \sim 1000$ steps) incurs prohibitive latency. Flow Matching (FM) (Lipman et al., 2022) offers an efficient alternative by learning a deterministic continuous normalizing flow (CNF) that directly transports samples from a simple prior to the data distribution. Compared with diffusion, FM is attractive for MVC because it uses far fewer integration steps, is deterministic under fixed noise and conditioning, avoids handcrafted noise schedules, and learns smoother OT-guided vector fields that are stable with coarse ODE solvers. The later runtime comparison in Table 4 and ODE-step study in Appendix Table 13 empirically validate this efficiency motivation.

Formally, FM learns a time-dependent vector field $v_\theta(\mathbf{z}_t, t)$ that defines an ordinary differential equation (ODE): $\frac{d\mathbf{z}_t}{dt} = v_\theta(\mathbf{z}_t, t)$. By integrating this ODE from $t = 0$ to $t = 1$, samples are transported from noise $\mathbf{z}_0 \sim \mathcal{N}(0, I)$ to data $\mathbf{z}_1 \sim p_{data}$. Following (Lipman et al., 2022), we adopt OT-guided affine paths with a small endpoint regularizer $\sigma_{\min} > 0$:

$$\mathbf{z}_t = \big(1 - (1 - \sigma_{\min})t\big)\mathbf{z}_0 + t\mathbf{z}_1, \qquad (3)$$

$$u_t = \mathbf{z}_1 - (1 - \sigma_{\min})\mathbf{z}_0. \qquad (4)$$

The network is trained to regress the target velocity $u_t$, yielding smooth trajectories that can be integrated with few Euler steps.

To leverage available semantic information, we extend FM to a conditional formulation: $v_\theta(\mathbf{z}_t, t, \mathbf{c})$, where $\mathbf{c}$ encodes context. In the aligned setting, the condition for sample $i$ is the consensus representation aggregated from its available

corresponding views, $\mathbf{c}_i = |\mathcal{V}_{\text{avail}}^{(i)}|^{-1} \sum_{v \in \mathcal{V}_{\text{avail}}^{(i)}} \mathbf{z}_i^{(v)}$. In the unaligned setting, this sample-wise cross-view averaging is mathematically invalid because index $i$ in different views may refer to different objects. We therefore condition each view independently on the cluster prototype: $\mathbf{c}_i^{(v)} = \boldsymbol{\mu}_{k^*}$, where $k^* = \arg\max_k q_{ik}^{(v)}$ is determined by the soft assignment in Eq. 6. This provides cluster-level semantic guidance for generation while cross-view consistency is enforced through $\mathcal{L}_{\text{gw}}$.

The training objective minimizes the discrepancy between the predicted and target vector fields:

$$\mathcal{L}_{\text{flow}} = \mathbb{E}_{t, \mathbf{z}_0, \mathbf{z}_1, \mathbf{c}} \left[ \|v_\theta(\mathbf{z}_t, t, \mathbf{c}) - u_t\|^2 \right] \qquad (5)$$

where $t \sim \mathcal{U}[0, 1]$, $\mathbf{z}_0 \sim \mathcal{N}(0, I)$, and $\mathbf{z}_1$ is the encoded latent of an observed view.

At test time, to impute a missing view for sample $i$, we: (1) sample initial noise $\mathbf{z}_0 \sim \mathcal{N}(0, I)$; (2) construct conditioning $\mathbf{c}$ based on the scenario; (3) integrate the ODE $\frac{d\mathbf{z}_t}{dt} = v_\theta(\mathbf{z}_t, t, \mathbf{c})$ using Euler method with 10 steps. The resulting $\mathbf{z}_1$ serves as the imputed latent representation, enabling complete multi-view fusion and clustering.

### 3.5. Soft Clustering

We employ soft cluster assignments based on cosine similarity between latent representations $\mathbf{z}_i$ and learnable centroids $\{\boldsymbol{\mu}_k\}_{k=1}^K$:

$$q_{ik} = \frac{\exp(\tilde{\mathbf{z}}_i^\top \tilde{\boldsymbol{\mu}}_k / \tau)}{\sum_{j=1}^K \exp(\tilde{\mathbf{z}}_i^\top \tilde{\boldsymbol{\mu}}_j / \tau)} \qquad (6)$$

where $\tilde{\mathbf{z}}_i$ and $\tilde{\boldsymbol{\mu}}_k$ are $\ell_2$-normalized. Following the DEC paradigm (Xie et al., 2016), we define a sharpened target distribution $P$ and minimize $\mathcal{L}_{\text{clu}} = KL(P\|Q)$ to encourage high-confidence assignments. The centroids are first initialized by K-Means after encoder-decoder warm-up, and are then treated as learnable parameters updated by gradient descent through $\mathcal{L}_{\text{clu}}$; during joint training, they are refined by K-Means every $T_{\text{update}} = 5$ epochs to keep them aligned with the evolving latent space.

### 3.6. Overall Objective

The total objective integrates five loss terms:

$$\mathcal{L}_{tot} = \mathcal{L}_{\text{flow}} + \lambda_g \mathcal{L}_{\text{gw}} + \lambda_c \mathcal{L}_{\text{clu}} + \lambda_r \mathcal{L}_{\text{rec}} + \lambda_t \mathcal{L}_{\text{con}} \quad (7)$$

where $\mathcal{L}_{\text{flow}}$ is the conditional flow matching loss in Eq. 5, $\mathcal{L}_{\text{gw}}$ is the structural alignment loss in Eq. 2, $\mathcal{L}_{\text{clu}}$ is the DEC-style clustering loss based on Eq. 6, and $\mathcal{L}_{\text{rec}} = \sum_v \|\mathbf{X}^{(v)} - g_{\psi_v}(\mathbf{z}^{(v)})\|^2$ preserves view-specific information. $\mathcal{L}_{\text{con}}$ is an InfoNCE loss that enhances instance-level cross-view consistency only when correspondences are known. For a pair of aligned views $(u, v)$, it is

$$\mathcal{L}_{\text{con}}^{u,v} = -\frac{1}{B} \sum_{i=1}^{B} \log \frac{\exp(\text{sim}(\mathbf{z}_i^{(u)}, \mathbf{z}_i^{(v)})/\tau_{\text{con}})}{\sum_{j=1}^{B} \exp(\text{sim}(\mathbf{z}_i^{(u)}, \mathbf{z}_j^{(v)})/\tau_{\text{con}})}, \quad (8)$$

where $\text{sim}(\cdot, \cdot)$ is cosine similarity and $\tau_{\text{con}}$ is the contrastive temperature. Positives are same-sample cross-view pairs and negatives are other samples in the mini-batch; $\mathcal{L}_{\text{con}}$ averages Eq. 8 over view pairs. For UMVC, such positives are unavailable, so we set $\lambda_t = 0$ and rely on $\mathcal{L}_{\text{gw}}$ for structure-level consistency. Default hyperparameters are $\lambda_g = 0.2$, $\lambda_c = 1.0$, $\lambda_r = 0.5$, and $\lambda_t = 0.3$ for aligned data or 0 for unaligned data. See Table 6 in the appendix for details.

### 3.7. Optimization

A three-phase training strategy is adopted: (1) *Warm-up I*: the encoder-decoder is trained with $\mathcal{L}_{\text{rec}}$ to establish a stable latent manifold; (2) *Warm-up II*: centroids are initialized via K-Means and single-view DEC loss is applied to develop cluster-separable representations; (3) *Joint Training*: end-to-end optimization with $\mathcal{L}_{tot}$, with centroids refined via K-Means every $T_{\text{update}} = 5$ epochs.

For implementation, we use Adam optimizer ($\text{lr} = 10^{-3}$), gradient clipping (max norm 1.0), and Euler ODE solver with 10 steps. The complete training procedure is provided in Algorithm 1 (Appendix).

## 4. Experiments

### 4.1. Experimental Setup

Experiments draw from six benchmark datasets: BDGP, COIL-20, CUB, Handwritten, NUS-WIDE, and Scene-15, summarized in Table 8 (Appendix E). Due to main-text space and baseline availability, Tables 1 and 2 report four representative datasets (BDGP, COIL-20, CUB, and NUS-WIDE). The appendix provides six-dataset robustness, ablation, sensitivity, convergence, visualization, and runtime evidence where the corresponding results are available. The incomplete and unaligned stress tests are evaluated separately in this version; mixed severe missingness-plus-misalignment evaluation is left as future work in the limitations.

OPTION is compared against state-of-the-art incomplete and unaligned MVC methods, including COMPLETER (Lin et al., 2021), SURE (Yang et al., 2023a), DealMVC (Yang et al., 2023b), the diffusion-based DCG (Zhang et al., 2025), MRG-UMC (Xin et al., 2025), CANDY (Guo et al., 2024), MFLVC (Xu et al., 2022), GCFAggMVC (Yan et al., 2023), FreeCSL (Dai et al., 2025), ROLL (Sun et al., 2025b) and PROTOCOL (Xue et al., 2025).

OPTION is implemented in PyTorch. We use an MLP-based encoder-decoder and a conditional flow matching network. The ODE solver uses the Euler method with $N = 10$ steps. The hyperparameters $\lambda_g$, $\lambda_c$, $\lambda_r$, and $\lambda_t$ are tuned via grid search around the default ranges reported in Appendix I. All experiments are run on a single NVIDIA RTX 3090 GPU. Evaluation metrics include Accuracy (ACC), Normalized Mutual Information (NMI), and Adjusted Rand Index (ARI). For readability, the main robustness tables report mean values, while Appendix G reports the corresponding mean±std results. OPTION robustness results are averaged over five independent runs with different random seeds. Ablation results are averaged over three random seeds because each component-removal variant requires a separate training run; sensitivity and convergence analyses use five random seeds.

### 4.2. Clustering Performance

We evaluate OPTION under both ideal (complete and aligned) and challenging (incomplete or unaligned) multi-view scenarios to comprehensively assess its robustness and effectiveness.

#### 4.2.1. INCOMPLETE MULTI-VIEW CLUSTERING

To simulate missing views, each view of each sample is independently removed with probability $r_m \in \{0\%, 10\%, 30\%, 50\%, 70\%\}$ while ensuring that every sample retains at least one observed view. Table 1 reports mean performance on the four representative main-text datasets;

the corresponding six-dataset mean±std results are provided in Appendix G.

Table 1 yields two key observations. First, OPTION achieves state-of-the-art performance in high-missingness scenarios. On CUB with 70% of views missing, it reaches 74.7% clustering accuracy, outperforming the second-best SURE by 27.7% and the diffusion-based DCG by 56.5%. This improvement can be attributed to conditional flow matching, which generates semantically coherent latent representations even from incomplete inputs. Second, OPTION exhibits strong stability across missing rates. While baseline methods degrade sharply—for example, DCG drops by 28.3% on COIL-20—OPTION maintains a nearly flat performance curve, with only a 2.6% decline. These results indicate that the learned transport paths effectively capture intrinsic cross-view correlations, enabling robust multi-view fusion under severe incompleteness.

### 4.2.2. UNALIGNED MULTI-VIEW CLUSTERING

For unaligned scenarios, we randomly select $r_u \times N$ samples in one view and shuffle their indices, where $r_u \in \{0\%, 20\%, 40\%, 60\%\}$; the remaining $(1 - r_u) \times N$ samples keep their original correspondences. Table 2 reports mean performance under this protocol. The corresponding six-dataset mean±std results are provided in Appendix G.

Table 2 shows that OPTION maintains high performance under misalignment, whereas baselines deteriorate significantly. For example, on BDGP (60% shuffle), OPTION achieves 46.3% accuracy compared to MFLVC's random-guess level (20.0%), and outperforms the robust baseline MRG-UMC by nearly 10%. This validates the role of our GW-inspired alignment: by enforcing consistency in intra-view geometric topology rather than relying on brittle index correspondences, OPTION supports robust fusion even when sample alignment is partially lost. The slight differences between the 0% columns of Tables 1 and 2 come from different random seeds and setting-specific tuning pipelines.

### 4.3. Ablation Study

To verify the effectiveness of each component, we conduct ablation studies on Scene-15 in an aligned incomplete setting with missing rate 0.7. This setting keeps valid same-sample positives for the contrastive term, while still testing the imputation and structural regularization components under severe missingness. Table 3 reports the results averaged over three random seeds. In fully unaligned UMVC, $\mathcal{L}_{con}$ is disabled by setting $\lambda_t = 0$.

Table 3 reveals several important findings. (1) Removing $\mathcal{L}_{con}$ or $\mathcal{L}_{clu}$ causes the largest performance drops (ACC drops by 12.2% and 11.0%, respectively), demonstrating that instance-level consistency and clustering supervision

are critical for discriminative representations when aligned positives are available. (2) Removing $\mathcal{L}_{rec}$ degrades ACC by 5.1%, confirming reconstruction loss helps preserve view-specific information. (3) Removing $\mathcal{L}_{gw}$ or $\mathcal{L}_{flow}$ leads to moderate drops (ACC drops by 1.1–1.2%), validating their roles in structural regularization and missing-view imputation under challenging conditions. Additional six-dataset ablation results, including the OT-coupling ablation when available, are reported in Appendix H.

### 4.4. Sensitivity Analysis

We investigate the robustness of OPTION to hyperparameter variations by analyzing nine hyperparameters on Scene-15, varying each across a wide range while fixing others at default values. Each configuration is evaluated over five random seeds. We quantify parameter sensitivity using the normalized performance range: $S = (\text{ACC}_{max} - \text{ACC}_{min})/\text{ACC}_{max}$, where lower values indicate greater stability.

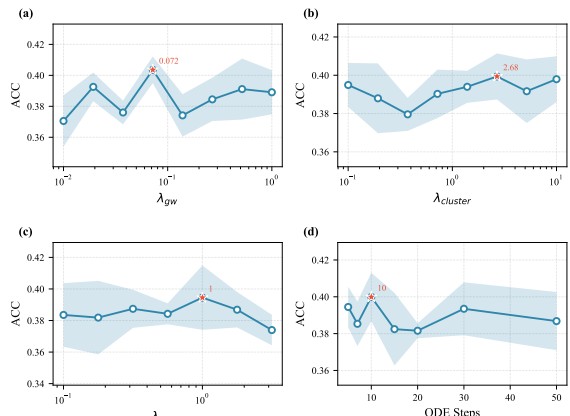

*Figure 2.* Sensitivity analysis of four key hyperparameters on Scene-15 (mean ± std, 5 runs).

Figure 2 reveals several key findings. (1) Overall robustness on Scene-15: all nine hyperparameters exhibit low sensitivity scores ($S < 0.1$), indicating that OPTION maintains stable clustering performance across a wide range of configurations on this dataset. Even $\lambda_g$, which shows the largest variation ($S = 0.082$), only causes $< 9\%$ relative performance change when varied over two orders of magnitude. (2) Efficient ODE integration: performance remains stable across a broad range of ODE steps, and $N = 10$ provides a robust quality-efficiency default for few-step generation. This claim is consistent with the measured runtime comparison in Table 4, and the explicit six-dataset ODE-step comparison in Appendix Table 13. (3) Loss weight stability: $\lambda_c$ and $\lambda_r$ show minimal sensitivity ($S \approx 0.05$), confirming that OPTION tolerates a broad range of loss weight configurations without significant performance degradation. (4) Architecture insensitivity: latent dimension, flow hidden dimension, learning rate, and dropout all show very

*Table 1.* Mean clustering performance (ACC, NMI, ARI) under varying missing rates on four representative main-text datasets. Complete six-dataset mean±std results are reported in Appendix G. Best results are **bolded** and second best are underlined.

| | METHOD | 0% | | | 10% | | | 30% | | | 50% | | | 70% | | |
|---|---|---|---|---|---|---|---|---|---|---|---|---|---|---|---|---|
| | | ACC | NMI | ARI | ACC | NMI | ARI | ACC | NMI | ARI | ACC | NMI | ARI | ACC | NMI | ARI |
| BDGP | OPTION | **50.5** | 27.3 | **24.8** | **50.4** | 27.2 | **24.7** | **50.4** | 27.2 | **24.7** | **46.6** | **19.8** | **17.8** | **33.9** | 9.3 | **6.1** |
| | MFLVC (CVPR22) | 20.0 | 0.0 | 0.0 | 20.0 | 0.0 | 0.0 | 20.0 | 0.0 | 0.0 | 26.8 | 2.9 | 1.7 | 27.6 | 4.9 | 2.5 |
| | SURE (TPAMI22) | 35.5 | 16.2 | 8.5 | 41.8 | 16.8 | 9.9 | 41.8 | 16.8 | 9.9 | 31.0 | 9.0 | 6.3 | 26.6 | 4.5 | 2.5 |
| | DEALMVC (CVPR23) | 24.8 | 4.8 | 0.6 | 24.8 | 4.8 | 0.6 | 24.8 | 4.8 | 0.6 | 24.5 | 2.4 | 1.3 | 23.0 | 1.2 | 0.4 |
| | GCFAGGMVC (CVPR23) | 31.1 | 5.9 | 4.8 | 31.1 | 5.9 | 4.8 | 31.1 | 5.9 | 4.8 | 24.5 | 2.4 | 1.3 | 26.3 | 2.5 | 1.4 |
| | DCG (AAAI25) | 45.8 | **27.9** | 21.7 | 45.8 | **27.9** | 21.7 | 45.8 | **27.9** | 21.7 | 39.2 | 18.3 | 11.3 | 27.6 | **9.7** | 1.5 |
| | FREECSL (CVPR25) | 31.2 | 6.0 | 4.9 | 31.2 | 6.0 | 4.9 | 31.2 | 6.0 | 4.9 | 25.6 | 2.9 | 1.5 | 25.3 | 1.5 | 0.8 |
| | ROLL (CVPR25) | 30.4 | 7.3 | 5.9 | 30.4 | 7.3 | 5.9 | 30.4 | 7.3 | 5.9 | 22.8 | 0.7 | 0.2 | 26.5 | 6.2 | 1.1 |
| | PROTOCOL (ICML25) | 35.4 | 10.6 | 9.3 | 35.4 | 10.6 | 9.3 | 35.4 | 10.6 | 9.3 | 25.4 | 2.7 | 1.6 | 26.5 | 2.9 | 1.2 |
| COIL-20 | OPTION | **88.7** | **93.8** | **86.4** | **88.4** | **93.7** | **86.2** | **88.4** | **93.7** | **86.2** | **87.6** | **92.9** | **84.6** | **86.1** | **92.3** | **83.3** |
| | MFLVC (CVPR22) | 57.9 | 77.0 | 48.9 | 57.9 | 77.0 | 48.9 | 57.9 | 77.0 | 48.9 | 48.3 | 63.3 | 35.1 | 37.9 | 54.5 | 24.4 |
| | SURE (TPAMI22) | 75.2 | 86.0 | 69.0 | 70.2 | 85.0 | 67.1 | 70.2 | 85.0 | 67.1 | 41.5 | 58.4 | 27.4 | 35.0 | 48.3 | 10.1 |
| | DEALMVC (CVPR23) | 57.7 | 83.0 | 60.4 | 57.7 | 83.0 | 60.4 | 57.7 | 83.0 | 60.4 | 34.6 | 55.9 | 19.3 | 29.4 | 38.5 | 9.5 |
| | GCFAGGMVC (CVPR23) | 78.5 | 87.2 | 73.1 | 78.5 | 87.2 | 73.1 | 78.5 | 87.2 | 73.1 | 41.9 | 58.1 | 28.7 | 35.0 | 50.5 | 19.3 |
| | DCG (AAAI25) | 62.7 | 82.8 | 58.4 | 62.7 | 82.8 | 58.4 | 62.7 | 82.8 | 58.4 | 47.7 | 70.2 | 36.8 | 34.4 | 54.4 | 19.8 |
| | FREECSL (CVPR25) | 39.2 | 54.9 | 23.3 | 39.2 | 54.9 | 23.3 | 39.2 | 54.9 | 23.3 | 29.6 | 40.2 | 13.0 | 27.5 | 40.4 | 11.9 |
| | ROLL (CVPR25) | 80.0 | 88.5 | 76.4 | 80.0 | 88.5 | 76.4 | 80.0 | 88.5 | 76.4 | 37.5 | 57.7 | 22.8 | 35.2 | 53.4 | 20.1 |
| | PROTOCOL (ICML25) | 63.7 | 82.6 | 61.9 | 63.7 | 82.6 | 61.9 | 63.7 | 82.6 | 61.9 | 41.0 | 60.4 | 29.1 | 31.9 | 49.6 | 18.0 |
| CUB | OPTION | **78.6** | **77.5** | **66.4** | **79.6** | **76.3** | **65.4** | **79.3** | **77.8** | **67.5** | **73.4** | **69.9** | **57.0** | **74.7** | **71.2** | **59.1** |
| | MFLVC (CVPR22) | 60.0 | 67.4 | 52.7 | 60.0 | 67.4 | 52.7 | 60.0 | 67.4 | 52.7 | 22.3 | 13.1 | 4.9 | 22.3 | 13.1 | 4.9 |
| | SURE (TPAMI22) | 70.7 | 61.9 | 52.3 | 75.7 | 67.1 | 56.3 | 75.7 | 67.1 | 56.3 | 47.0 | 49.9 | 28.7 | 47.0 | 49.9 | 28.7 |
| | DEALMVC (CVPR23) | 65.3 | 69.6 | 54.4 | 65.3 | 69.6 | 54.4 | 65.3 | 69.6 | 54.4 | 34.3 | 36.0 | 18.0 | 34.3 | 36.0 | 18.0 |
| | GCFAGGMVC (CVPR23) | 78.2 | 74.9 | 65.3 | 78.2 | 74.9 | 65.3 | 78.2 | 74.9 | 65.3 | 38.8 | 38.6 | 20.4 | 38.8 | 38.6 | 20.4 |
| | DCG (AAAI25) | 22.2 | 13.8 | 4.8 | 22.2 | 13.8 | 4.8 | 22.2 | 13.8 | 4.8 | 18.2 | 5.9 | 0.9 | 18.2 | 5.9 | 0.9 |
| | FREECSL (CVPR25) | 26.3 | 16.6 | 6.9 | 26.3 | 16.6 | 6.9 | 26.3 | 16.6 | 6.9 | 20.2 | 8.9 | 2.7 | 20.2 | 8.9 | 2.7 |
| | ROLL (CVPR25) | 75.0 | 72.8 | 59.6 | 75.0 | 72.8 | 59.6 | 75.0 | 72.8 | 59.6 | 38.3 | 41.3 | 22.0 | 38.3 | 41.3 | 22.0 |
| | PROTOCOL (ICML25) | 55.5 | 59.9 | 44.9 | 55.5 | 59.9 | 44.9 | 55.5 | 59.9 | 44.9 | 33.3 | 28.8 | 5.0 | 33.3 | 28.8 | 5.0 |
| NUS-WIDE | OPTION | **19.0** | 18.7 | 6.3 | **19.4** | 17.9 | 5.2 | **19.9** | 15.9 | **5.9** | **19.1** | 17.3 | **6.5** | **17.9** | 20.2 | **6.8** |
| | MFLVC (CVPR22) | 15.6 | 23.1 | 6.2 | 15.6 | 23.1 | 6.2 | 15.2 | **21.5** | 5.3 | 14.9 | **19.7** | 4.9 | 12.5 | 15.6 | 3.0 |
| | SURE (TPAMI22) | 14.1 | 17.2 | 3.6 | 13.4 | 17.5 | 3.7 | 11.9 | 14.3 | 2.1 | 11.2 | 12.8 | 1.4 | 11.9 | 13.0 | 0.8 |
| | DEALMVC (CVPR23) | 16.4 | 20.8 | 6.3 | 16.4 | 20.8 | 6.3 | 12.2 | 7.7 | 0.4 | 10.2 | 10.0 | 0.7 | 9.8 | 8.5 | 0.3 |
| | GCFAGGMVC (CVPR23) | 17.6 | **23.2** | **6.9** | 17.6 | **23.2** | **6.9** | 12.2 | 15.1 | 2.7 | 9.8 | 12.1 | 1.4 | 9.3 | 11.8 | 1.3 |
| | DCG (AAAI25) | 17.1 | 20.3 | 5.1 | 17.1 | 20.3 | 5.1 | 14.2 | 18.1 | 3.8 | 13.2 | 16.1 | 2.7 | 13.5 | 14.4 | 2.7 |
| | FREECSL (CVPR25) | 10.1 | 10.5 | 1.2 | 10.1 | 10.5 | 1.2 | 9.6 | 9.9 | 0.7 | 9.2 | 9.5 | 0.6 | 8.4 | 8.4 | 0.3 |
| | ROLL (CVPR25) | 17.2 | 21.3 | 5.7 | 17.2 | 21.3 | 5.7 | 12.0 | 15.9 | 3.0 | 9.8 | 11.9 | 1.4 | 10.0 | 11.8 | 1.3 |
| | PROTOCOL (ICML25) | 15.5 | 17.8 | 3.7 | 15.5 | 17.8 | 3.7 | 13.8 | 15.6 | 2.6 | 13.2 | 13.5 | 1.6 | 13.7 | 12.5 | 1.9 |

low sensitivity ($S < 0.05$), demonstrating that OPTION's performance stems from its formulation rather than architectural tuning.

These results have practical implications: the low sensitivity across all hyperparameters on Scene-15 suggests that OPTION can often be deployed with default settings before dataset-specific tuning. For practitioners, we recommend: $\lambda_g \in [0.05, 0.2]$, $N = 10$ ODE steps, $\lambda_c \in [1.0, 3.0]$, $\lambda_r \in [0.5, 1.5]$, and $\lambda_t = 0.3$ for aligned data or 0 for unaligned data. Complete Scene-15 results for all nine parameters are provided in Appendix I; additional cross-dataset sensitivity summaries are reported in Appendix I.4.

### 4.5. Convergence Analysis

To demonstrate the optimization stability and convergence properties of OPTION, we conduct a comprehensive training dynamics analysis with statistical validation. We run 5 independent experiments with different random seeds (42–46) and track the evolution of loss components and clustering metrics throughout training on the Handwritten dataset. Figure 3 illustrates the convergence behavior with mean values and standard deviation bands.

The multi-seed experiments demonstrate OPTION's robustness: all clustering metrics (ACC, NMI, ARI) converge to stable values across 5 independent runs, with the shaded regions in Figure 3 indicating consistently low variance. The relatively tight standard deviation bands suggest that OPTION's optimization is reproducible regardless of random

*Table 2.* Mean clustering performance (ACC, NMI, ARI) under varying unaligned rates on four representative main-text datasets. Complete six-dataset mean±std results are reported in Appendix G. Best results are **bolded** and second best are underlined.

| | METHOD | 0% ACC | NMI | ARI | 20% ACC | NMI | ARI | 40% ACC | NMI | ARI | 60% ACC | NMI | ARI |
|---|---|---|---|---|---|---|---|---|---|---|---|---|---|
| **BDGP** | OPTION | **50.3** | **27.6** | **25.1** | **48.3** | **25.2** | **22.8** | **48.0** | **25.3** | **23.1** | **46.3** | **23.6** | **21.7** |
| | MFLVC (CVPR22) | 20.0 | 0.0 | 0.0 | 20.0 | 0.0 | 0.0 | 20.0 | 0.0 | 0.0 | 20.0 | 0.0 | 0.0 |
| | GCFAGGMVC (CVPR23) | 31.1 | 5.9 | 4.8 | 29.6 | 4.2 | 3.4 | 27.2 | 2.6 | 1.7 | 25.5 | 1.8 | 1.1 |
| | MRG-UMC (TNNLS25) | 37.9 | 12.1 | 9.4 | 35.2 | 7.9 | 6.3 | 26.5 | 2.6 | 1.9 | 36.6 | 15.3 | 12.3 |
| | CANDY (NEURIPS24) | 33.4 | 7.3 | 5.6 | 28.5 | 3.9 | 3.2 | 26.2 | 2.0 | 1.5 | 25.3 | 1.6 | 1.0 |
| | FREECSL (CVPR25) | 31.2 | 6.0 | 4.9 | 29.3 | 4.1 | 3.3 | 26.4 | 2.0 | 1.5 | 25.5 | 1.7 | 1.0 |
| | ROLL (CVPR25) | 30.4 | 7.3 | 5.9 | 29.0 | 5.9 | 4.8 | 26.8 | 2.7 | 2.0 | 25.5 | 1.7 | 1.4 |
| | PROTOCOL (ICML25) | 35.4 | 10.6 | 9.3 | 31.3 | 4.8 | 3.3 | 29.2 | 2.6 | 1.6 | 26.4 | 1.5 | 0.9 |
| **COIL-20** | OPTION | **88.7** | **93.8** | **86.4** | **73.5** | **75.5** | **60.3** | **66.4** | **63.2** | **45.8** | **51.5** | **49.2** | **28.4** |
| | MFLVC (CVPR22) | 57.9 | 77.0 | 48.9 | 44.6 | 58.2 | 34.6 | 38.3 | 44.9 | 22.0 | 33.1 | 36.9 | 15.5 |
| | GCFAGGMVC (CVPR23) | 78.5 | 87.2 | 73.1 | 61.9 | 67.6 | 47.3 | 51.2 | 56.0 | 33.8 | 38.3 | 44.3 | 20.3 |
| | MRG-UMC (TNNLS25) | 61.3 | 83.1 | 59.3 | 59.2 | 67.7 | 44.4 | 39.6 | 51.4 | 23.9 | 33.3 | 43.1 | 16.8 |
| | CANDY (NEURIPS24) | 47.7 | 64.3 | 34.4 | 46.0 | 55.0 | 30.1 | 40.6 | 47.4 | 23.4 | 35.8 | 39.0 | 16.0 |
| | FREECSL (CVPR25) | 39.2 | 54.9 | 23.3 | 33.3 | 43.1 | 15.4 | 27.1 | 33.6 | 9.5 | 22.9 | 25.4 | 4.9 |
| | ROLL (CVPR25) | 80.0 | 88.5 | 76.4 | 64.6 | 71.4 | 51.5 | 53.1 | 56.1 | 33.1 | 41.7 | 48.2 | 24.2 |
| | PROTOCOL (ICML25) | 63.7 | 82.6 | 61.9 | 46.7 | 63.5 | 37.7 | 44.0 | 52.8 | 27.6 | 33.3 | 41.7 | 15.9 |
| **CUB** | OPTION | **79.2** | **77.4** | **67.1** | **69.5** | **61.9** | **51.1** | **60.5** | 51.6 | 39.4 | 53.0 | 48.5 | 35.9 |
| | MFLVC (CVPR22) | 60.0 | 67.4 | 52.7 | 50.0 | 54.9 | 41.3 | 36.2 | 36.1 | 23.7 | 40.8 | 31.8 | 22.5 |
| | GCFAGGMVC (CVPR23) | 78.2 | 74.9 | 65.3 | 58.3 | 49.5 | 38.1 | 48.0 | 29.9 | 21.3 | 39.0 | 20.9 | 13.1 |
| | MRG-UMC (TNNLS25) | 78.5 | 74.6 | 63.8 | 63.2 | 53.3 | 41.6 | 45.2 | 33.0 | 22.9 | 33.2 | 21.3 | 12.1 |
| | CANDY (NEURIPS24) | 60.7 | 51.5 | 38.9 | 45.5 | 32.9 | 21.5 | 32.5 | 15.9 | 9.2 | 27.8 | 11.7 | 5.6 |
| | FREECSL (CVPR25) | 26.3 | 16.6 | 6.9 | 23.3 | 11.8 | 4.3 | 23.5 | 9.2 | 3.6 | 16.8 | 4.9 | 0.7 |
| | ROLL (CVPR25) | 75.0 | 72.8 | 59.6 | 52.5 | 49.0 | 37.4 | 41.8 | 24.4 | 15.7 | 35.3 | 17.8 | 10.2 |
| | PROTOCOL (ICML25) | 55.5 | 59.9 | 44.9 | 54.3 | 60.1 | 44.0 | 59.3 | **61.3** | **46.3** | **55.7** | **61.7** | **46.7** |
| **NUS-WIDE** | OPTION | **19.6** | 16.0 | 5.5 | **17.8** | 15.9 | **5.2** | **15.9** | 12.2 | **3.2** | **14.1** | 9.5 | 1.7 |
| | MFLVC (CVPR22) | 15.6 | 23.1 | 6.2 | 15.1 | 18.6 | 4.4 | 13.0 | 15.3 | 3.2 | 11.2 | 11.7 | 1.7 |
| | GCFAGGMVC (CVPR23) | 17.6 | **23.2** | **6.9** | 14.3 | 18.2 | 4.3 | 13.5 | **15.3** | 3.2 | 11.0 | 11.4 | 1.6 |
| | MRG-UMC (TNNLS25) | 17.8 | 19.1 | 4.8 | 15.2 | 15.2 | 3.3 | 13.3 | 13.9 | 2.2 | 12.4 | 12.0 | 1.6 |
| | CANDY (NEURIPS24) | 17.5 | 22.7 | 6.9 | 14.6 | **18.9** | 4.7 | 13.6 | 15.3 | 3.2 | 11.8 | 12.6 | 1.8 |
| | FREECSL (CVPR25) | 10.1 | 10.5 | 1.2 | 9.2 | 10.0 | 0.8 | 9.0 | 9.1 | 0.5 | 8.9 | 8.2 | 0.3 |
| | ROLL (CVPR25) | 17.2 | 21.3 | 5.7 | 14.3 | 18.2 | 3.9 | 13.2 | 14.6 | 2.9 | 12.2 | **13.1** | **1.9** |
| | PROTOCOL (ICML25) | 15.5 | 17.8 | 3.7 | 12.3 | 13.6 | 2.1 | 11.2 | 10.9 | 1.3 | 10.5 | 8.9 | 0.5 |

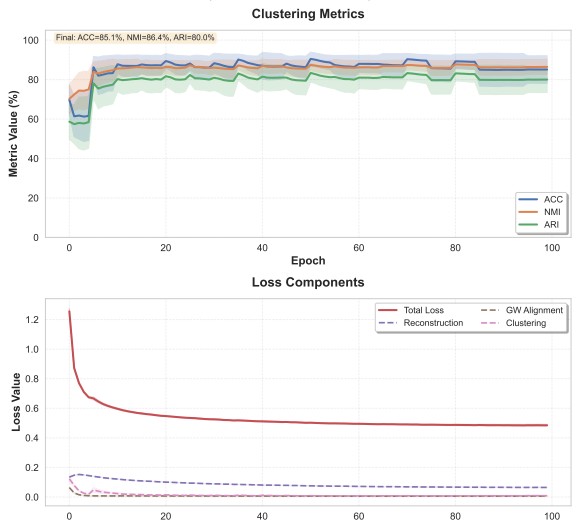

*Figure 3.* Convergence analysis on Handwritten (mean ± std, 5 runs). Top: clustering metrics; Bottom: loss components.

*Table 3.* Ablation study on Scene-15. Best results are in **bold**.

| | Components | | | | | Scene-15 | | |
|---|---|---|---|---|---|---|---|---|
| Variant | $\mathcal{L}_{\text{gw}}$ | $\mathcal{L}_{\text{flow}}$ | $\mathcal{L}_{\text{con}}$ | $\mathcal{L}_{\text{clu}}$ | $\mathcal{L}_{\text{rec}}$ | ACC | NMI | ARI |
| w/o GW | × | ✓ | ✓ | ✓ | ✓ | 36.4±0.2 | 35.0±0.6 | 21.3±1.0 |
| w/o Flow | ✓ | × | ✓ | ✓ | ✓ | 36.3±1.0 | 37.0±1.1 | 22.4±1.4 |
| w/o Contrastive | ✓ | ✓ | × | ✓ | ✓ | 25.3±1.5 | 24.9±0.8 | 11.2±1.4 |
| w/o Clustering | ✓ | ✓ | ✓ | × | ✓ | 26.5±1.4 | 25.1±0.7 | 12.5±0.4 |
| w/o Recon | ✓ | ✓ | ✓ | ✓ | × | 32.4±0.5 | 31.2±0.3 | 17.2±0.2 |
| Full model | ✓ | ✓ | ✓ | ✓ | ✓ | **37.5±0.4** | **37.1±0.4** | **23.2±0.0** |

initialization, a desirable property for practical deployment.

The bottom panel of Figure 3 shows the evolution of individual loss components. All losses decrease monotonically with small variance bands, indicating: (i) no single objective dominates the optimization; (ii) the multi-task learning is well-balanced across reconstruction ($\mathcal{L}_{\text{rec}}$), GW-inspired alignment ($\mathcal{L}_{\text{gw}}$), and clustering ($\mathcal{L}_{\text{clu}}$) objectives. The tight variance bands confirm that OPTION's optimization landscape is well-conditioned across different random seeds.

The top panel of Figure 3 reveals several favorable optimization properties: (i) Fast convergence: all three clustering

metrics (ACC, NMI, ARI) show rapid initial improvement in the first 20 epochs, then gradually stabilize. (ii) Smooth trajectories: the mean curves exhibit monotonic improvement with minimal oscillation, suggesting stable gradient flow facilitated by the optimal transport formulation. (iii) Consistent variance: the standard deviation bands remain relatively constant throughout training, indicating that convergence behavior is reproducible across different random seeds. Complete six-dataset convergence results are provided in Appendix J.

## 4.6. Visualization

We visualize latent representations with t-SNE at the beginning and end of training on two representative datasets. Figure 4 reports the main-text visualization, and the appendix provides complete six-dataset visualizations in Appendix K.

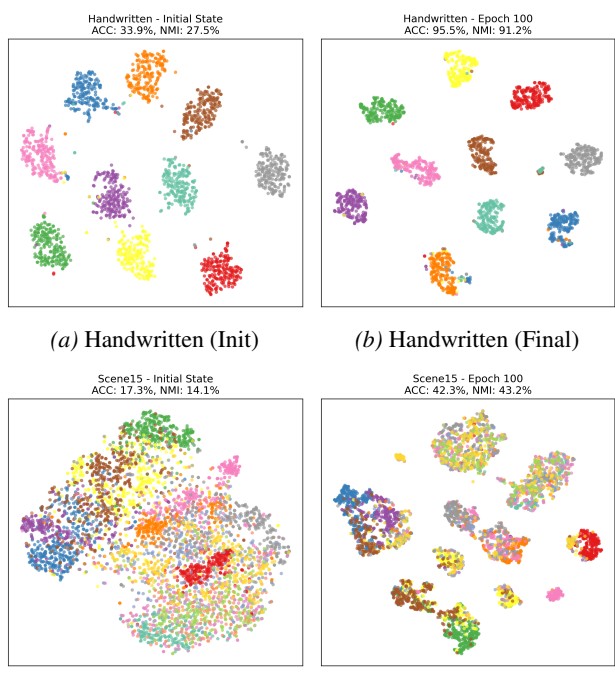

*(a)* Handwritten (Init)      *(b)* Handwritten (Final)

*(c)* Scene-15 (Init)      *(d)* Scene-15 (Final)

*Figure 4.* t-SNE visualization of latent representations before and after training.

Initially (Epoch 0), representations are randomly distributed with heavily overlapping clusters. After training, clusters become compact and well-separated: Handwritten achieves 95.5% accuracy (from 33.9%), and Scene-15 reaches 42.3% (from 17.3%).

## 4.7. Computational Efficiency Analysis

We evaluate computational efficiency through sampling/imputation latency against the diffusion-based DCG baseline and wall-clock runtime against non-diffusion ones. All experiments use the same batch size (256) and identical hardware (NVIDIA RTX 3090).

*Table 4.* Inference speed comparison (ms per batch). OPTION uses 10 ODE steps; DCG uses 100 diffusion steps.

| Dataset | OPTION (ms) | DCG (ms) | Speedup |
|---|---|---|---|
| Scene-15 | 3.08 | 25.61 | 8.3× |
| Handwritten | 4.06 | 26.07 | 6.4× |
| COIL-20 | 3.81 | 26.50 | 7.0× |
| BDGP | 3.10 | 26.27 | 8.5× |
| CUB | 2.63 | 25.96 | 9.9× |
| NUS-WIDE | 3.00 | 25.29 | 8.4× |
| **Average** | **3.28** | **25.95** | **8.1×** |

Table 4 demonstrates that OPTION achieves $6.4\times$ to $9.9\times$ speedup over DCG (average $8.1\times$) due to: (1) OT-guided affine paths requiring only 10 ODE steps vs. 100 diffusion denoising iterations; (2) deterministic ODEs vs. stochastic sampling; (3) minimal fixed memory footprint across all datasets. Thus, the practical speed claim in this paper is the measured wall-clock gain in Table 4; the larger step-count reduction only describes the algorithmic gap between few-step flow matching and long-chain diffusion samplers. For practical deployment, this translates to approximately 3 ms per 256-sample batch on a single RTX 3090, enabling real-time clustering updates.

For broader context beyond diffusion-based baselines, Table 5 reports wall-clock runtime against representative non-diffusion methods under the same benchmarking protocol across all six datasets. On NUS-WIDE, OPTION remains faster than MRG-UMC, SURE, and CANDY, requiring 7716.2 ms compared with 24342.8 ms, 34493.1 ms, and 40109.0 ms, respectively.

*Table 5.* Runtime comparison with representative non-diffusion baselines (ms) under the same benchmarking protocol. Lower is better.

| Method | BDGP | COIL-20 | CUB | Handwritten | Scene-15 | NUS-WIDE |
|---|---|---|---|---|---|---|
| OPTION | 2862.9 | 6760.3 | 2449.9 | 3514.8 | 6585.7 | 7716.2 |
| MRG-UMC (TNNLS25) | 6765.4 | 13295.0 | 3964.0 | 18035.3 | 6111.8 | 24342.8 |
| CANDY (NeurIPS24) | 12899.6 | 16644.1 | 5687.9 | 44773.1 | 11957.8 | 40109.0 |
| SURE (TPAMI22) | 15843.2 | 37692.5 | 15358.6 | 14552.0 | 15038.1 | 34493.1 |

## 5. Conclusion

This paper introduced OPTION, a unified framework for incomplete and unaligned multi-view clustering that integrates OT-guided flow matching with GW-inspired structural alignment. By learning deterministic transport paths with only 10 ODE steps, OPTION achieves an average $8.1\times$ speedup over the diffusion-based baseline while maintaining strong imputation quality. The structural regularizer supports alignment-free multi-view fusion without hard correspondence assignment. Experiments on the evaluated ideal, incomplete, and unaligned scenarios show competitive clustering performance and robust hyperparameter stability.

## Acknowledgements

This work was supported in part by the National Natural Science Foundation of China (No. 62506116), the Hebei Natural Science Foundation (No. F2025205006), the Science Research Project of the Hebei Education Department (No. BJ2026004), and the Open Project of the Key Laboratory of Tibetan Information Processing, Ministry of Education, Qinghai Normal University (No. QHSF-CS-2603).

## Impact Statement

This paper presents work whose goal is to advance the field of machine learning. There are many potential societal consequences of our work, none of which we feel must be specifically highlighted here.

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

# A. Preliminaries on Flow Matching

Flow Matching (FM) (Lipman et al., 2022) provides a simulation-free framework for learning generative models by directly regressing a vector field that transports samples from a source distribution $p_0$ (typically Gaussian noise) to a target distribution $p_1$ (data). Unlike diffusion models that require iterative stochastic denoising, FM learns a deterministic continuous normalizing flow (CNF) via ordinary differential equations (ODEs).

**Continuous Normalizing Flow.** Given a time-dependent vector field $u_t : \mathbb{R}^d \times [0, 1] \to \mathbb{R}^d$, the flow is defined by the ODE:

$$\frac{d\mathbf{z}_t}{dt} = u_t(\mathbf{z}_t), \quad \mathbf{z}_0 \sim p_0, \quad t \in [0, 1] \tag{9}$$

Solving this ODE forward from $t = 0$ to $t = 1$ generates samples: $\mathbf{z}_1 = \mathbf{z}_0 + \int_0^1 u_t(\mathbf{z}_t)dt$. The distribution evolves according to the continuity equation $\frac{\partial p_t}{\partial t} + \nabla \cdot (p_t u_t) = 0$, ensuring $p_0$ is transported to $p_1$.

**Conditional Flow Matching.** Directly learning $u_t$ requires access to $p_t$ at all intermediate times, which is intractable. The Conditional Flow Matching (CFM) objective (Lipman et al., 2022) bypasses this by conditioning on data samples $\mathbf{z}_1 \sim p_1$. For each data point $\mathbf{z}_1$, we define a conditional probability path that interpolates between noise $\mathbf{z}_0 \sim \mathcal{N}(0, I)$ and $\mathbf{z}_1$. We use the same affine path as Eq. 4:

$$\mathbf{z}_t = \big(1 - (1 - \sigma_{\min})t\big)\mathbf{z}_0 + t\mathbf{z}_1, \tag{10}$$
$$u_t = \mathbf{z}_1 - (1 - \sigma_{\min})\mathbf{z}_0. \tag{11}$$

where $\sigma_{\min} > 0$ is a small regularization constant. The neural network $v_\theta(\mathbf{z}, t, \mathbf{c})$ is trained via:

$$\mathcal{L}_{\text{flow}}(\theta) = \mathbb{E}_{t, \mathbf{z}_1, \mathbf{z}_0, \mathbf{z}_t, \mathbf{c}} \|v_\theta(\mathbf{z}_t, t, \mathbf{c}) - u_t\|^2. \tag{12}$$

This enables efficient training without ODE simulation, and at inference, the learned vector field can be integrated using few-step ODE solvers (e.g., 10 Euler steps).

# B. Training Algorithm

Algorithm 1 presents the complete end-to-end training procedure for OPTION. The algorithm follows a three-phase strategy: (1) encoder-decoder pretraining with reconstruction loss to establish a stable latent manifold, (2) centroid initialization via K-Means and single-view DEC pretraining to develop cluster-separable representations, and (3) joint optimization with all loss components, where centroids receive gradient updates and are refreshed by K-Means every five epochs. A key distinction is made between aligned (IMVC) and unaligned (UMVC) scenarios: in the aligned setting, the flow conditioning uses consensus representations aggregated across views; in the unaligned setting, each view operates independently with cluster centroid conditioning, as cross-view sample correspondences are unknown.

# C. Loss Configuration

Table 6 summarizes the loss function configuration for different multi-view clustering scenarios. The key difference between aligned (IMVC) and unaligned (UMVC) settings lies in two aspects: (1) the flow conditioning source—consensus representation for IMVC versus cluster centroid for UMVC, and (2) the contrastive loss $\mathcal{L}_{\text{con}}$, which is only applicable when sample correspondences are known. All other loss components remain active in both scenarios, ensuring that OPTION maintains its core capabilities of generative imputation, structural regularization, and clustering supervision regardless of the alignment assumption.

# D. Practical Hyperparameter Defaults

Table 7 summarizes the default configuration used as a starting point in our experiments. For new datasets, we recommend first using these values and then tuning only the most influential weights, $\lambda_g$, $\lambda_c$, and $\lambda_r$, if validation stability is unsatisfactory. The contrastive weight is setting-dependent: it is enabled only when sample correspondences are known.

---

**Algorithm 1** End-to-End Training for OPTION

---

**Input:** Multi-view data $\mathcal{X}$, view mask $\mathbf{M}$, number of clusters $K$, hyperparameters $\lambda_g, \lambda_c, \lambda_r, \lambda_t$, centroid refresh interval $T_{\text{update}} = 5$, alignment flag `is_aligned`.
**Initialize:** encoder-decoder parameters $\phi, \psi$, vector-field parameters $\theta$.
**Phase I: Encoder-decoder warm-up.**
**for** epoch $= 1$ to $E_{\text{rec}}$ **do**
    **for** each mini-batch $\mathcal{B}$ **do**
        Encode observed views and reconstruct them.
        Update $\phi, \psi$ by minimizing $\mathcal{L}_{\text{rec}}$ on observed entries indicated by $\mathbf{M}$.
    **end for**
**end for**
Run K-Means on warm-up latent representations and initialize centroids $\{\boldsymbol{\mu}_k\}_{k=1}^{K}$.
**Phase II: Clustering warm-up.**
**for** epoch $= 1$ to $E_{\text{clu}}$ **do**
    **for** each mini-batch $\mathcal{B}$ **do**
        Compute soft assignments $Q$ and sharpened targets $P$.
        Update encoders and centroids by minimizing $\mathcal{L}_{\text{clu}} = KL(P_{\mathcal{B}} \| Q_{\mathcal{B}})$.
    **end for**
**end for**
**Phase III: Joint training.**
**for** epoch $= 1$ to $E_{\text{joint}}$ **do**
    **for** each mini-batch $\mathcal{B}$ **do**
        Encode observed views: $\mathbf{z}^{(v)} = f_{\phi_v}(\mathbf{x}^{(v)})$.
        **if** `is_aligned` **then**
            For each sample $i$, compute available-view consensus $\mathbf{c}_i = |\mathcal{V}_{\text{avail}}^{(i)}|^{-1} \sum_{v \in \mathcal{V}_{\text{avail}}^{(i)}} \mathbf{z}_i^{(v)}$.
            Enable $\mathcal{L}_{\text{con}}$ with same-sample cross-view positives.
        **else**
            For each view $v$ and sample $i$, compute $q_{ik}^{(v)}$ by Eq. 6.
            Set $k_i^{*(v)} = \arg\max_k q_{ik}^{(v)}$ and condition $\mathbf{c}_i^{(v)} = \boldsymbol{\mu}_{k_i^{*(v)}}$.
            Disable $\mathcal{L}_{\text{con}}$ by setting $\lambda_t = 0$.
        **end if**
        Sample noise $\mathbf{z}_0 \sim \mathcal{N}(0, I)$ and time $t \sim \mathcal{U}[0, 1]$.
        Construct $\mathbf{z}_t$ and target velocity by Eq. 4; compute $\mathcal{L}_{\text{flow}}$.
        Compute $\mathcal{L}_{\text{gw}}, \mathcal{L}_{\text{clu}}, \mathcal{L}_{\text{rec}}$, and, if aligned, $\mathcal{L}_{\text{con}}$.
        Update all parameters and centroids via Adam with gradient clipping using Eq. 7.
    **end for**
    **if** epoch mod $T_{\text{update}} = 0$ **then**
        Refine centroids $\{\boldsymbol{\mu}_k\}_{k=1}^{K}$ via K-Means on current embeddings.
    **end if**
**end for**
**Output:** Trained model, Cluster assignments $\arg\max_k q_{ik}$.

---

*Table 6.* Loss function configuration for different MVC scenarios.

| Component | Aligned (IMVC) | Unaligned (UMVC) |
|---|---|---|
| **Flow Conditioning** | Consensus $\mathbf{z}^{(c)}$ | Centroid $\boldsymbol{\mu}_{k^*}$ |
| $\mathcal{L}_{\text{flow}}$ (Generative) | ✓ | ✓ |
| $\mathcal{L}_{\text{gw}}$ | ✓ | ✓ |
| $\mathcal{L}_{\text{clu}}$ | ✓ | ✓ |
| $\mathcal{L}_{\text{rec}}$ | ✓ | ✓ |
| $\mathcal{L}_{\text{con}}$ | ✓ | ✗ |

The generative CFM loss is always active; only the conditioning source differs.

*Table 7.* Practical default hyperparameters and recommended tuning ranges.

| Hyperparameter | Default | Recommended range |
|---|---|---|
| $\lambda_g$ | 0.2 | [0.05, 0.2] |
| $\lambda_c$ | 1.0 | [1.0, 3.0] |
| $\lambda_r$ | 0.5 | [0.5, 1.5] |
| $\lambda_t$ | 0.3 / 0 | 0.3 for aligned, 0 for unaligned |
| ODE steps $N$ | 10 | [5, 20] |
| Learning rate | $10^{-3}$ | $[10^{-4}, 10^{-2}]$ |
| Latent dimension | 128 | [32, 384] |
| Dropout | 0.1 | [0, 0.5] |

## E. Dataset Details

To comprehensively evaluate the performance of OPTION, we conducted experiments on six widely used multi-view benchmark datasets covering various domains, including generic object recognition, digit classification, biological data analysis, and scene categorization. Table 8 summarizes these datasets. Detailed descriptions are provided below.

*Table 8.* Statistical summary of the benchmark multi-view datasets used in experiments. $N$: Sample size, $V$: Number of views, $K$: Number of clusters/categories. The View Dimensions column lists the feature dimensionality for each specific view.

| DATASET | $N$ | $V$ | $K$ | VIEW DIMENSIONS |
|---|---|---|---|---|
| BDGP | 2,500 | 2 | 5 | [1750, 79] |
| COIL-20 | 1,440 | 3 | 20 | [1024, 3304, 6750] |
| CUB | 11,788 | 2 | 200 | [1024, 300] |
| HANDWRITTEN | 2,000 | 6 | 10 | [240, 76, 216, 47, 64, 6] |
| NUS-WIDE | 30,000 | 5 | 31 | [64, 144, 73, 128, 225] |
| SCENE-15 | 4,485 | 3 | 15 | [20, 59, 40] |

**BDGP (Berkeley Drosophila Genome Project).** This is a biological dataset consisting of 2,500 samples belonging to 5 categories of Drosophila embryos. It contains two heterogeneous views: the first view consists of 1,750-dimensional visual features extracted from lateral and dorsal images of the embryos, and the second view comprises 79-dimensional textual features derived from the corresponding gene expression terms. This dataset challenges the model to align visual patterns with semantic biological descriptions.

**COIL-20 (Columbia Object Image Library).** This dataset contains 1,440 grayscale images of 20 distinct objects (e.g., ducks, cars, cups). Each object was rotated on a turntable through 360 degrees to capture 72 poses. Three views are constructed to represent different visual characteristics: intensity features, LBP (Local Binary Patterns) features, and Gabor features. The dataset tests the model's ability to recognize 3D objects from multi-modal 2D feature descriptors.

**CUB (Caltech-UCSD Birds-200-2011).** This is a fine-grained image classification dataset containing 11,788 images of 200 bird species. We utilize a multi-view setting composed of deep visual features and textual features. View 1 typically consists of 1,024-dimensional features extracted via a pre-trained deep neural network (e.g., GoogleNet), capturing high-level visual semantics. View 2 consists of 300-dimensional textual features (e.g., Doc2Vec) obtained from the text descriptions associated with each bird class. The large number of categories and fine-grained differences make this a challenging benchmark.

**Handwritten.** This classic dataset consists of 2,000 samples of handwritten numerals from '0' to '9' (10 classes), with 200 samples per class. It provides a rich multi-view representation using six different feature descriptors: Pixel averages (240-dim), Fourier coefficients (76-dim), Profile correlations (216-dim), Zernike moments (47-dim), Karhunen-Love coefficients (64-dim), and Morphological features (6-dim). The high number of views ($V = 6$) makes it suitable for evaluating the scalability of multi-view fusion mechanisms.

**NUS-WIDE.** This is a large-scale real-world web image dataset collected from Flickr. Following standard protocols in multi-view clustering literature, we use a subset of 30,000 images belonging to 31 frequent object categories. The dataset includes five low-level visual feature views: 64-D Color Histogram, 144-D Color Correlogram, 73-D Edge Direction Histogram, 128-D Wavelet Texture, and 225-D Block-wise Color Moments. This dataset evaluates the model's performance on large-scale, diverse, and potentially noisy web data.

**Scene-15.** This dataset contains 4,485 images classifying 15 distinct indoor and outdoor scene categories (e.g., bedroom, kitchen, coast, forest). The multi-view features are extracted to capture different aspects of the scene environment.

Common views include 20-dimensional GIST features (spatial envelope), 59-dimensional PHOG (Pyramid Histogram of Oriented Gradients) features, and 40-dimensional LBP (Local Binary Patterns) features. It serves as a benchmark for scene understanding and structural feature integration.

## F. Experimental Protocol

**Incomplete MVC.** For a missing rate $r_m$, each view of each sample is independently removed with probability $r_m \in \{0\%, 10\%, 30\%, 50\%, 70\%\}$, while enforcing that every sample retains at least one observed view. Reconstruction losses are computed only on observed entries, and missing latents are generated by the conditional flow module.

**Unaligned MVC.** For an unaligned rate $r_u$, we randomly select $r_u \times N$ samples in one view and shuffle their indices, with $r_u \in \{0\%, 20\%, 40\%, 60\%\}$. The remaining $(1 - r_u) \times N$ samples keep their original correspondences. In this setting, sample-wise cross-view averaging and contrastive positives are disabled; flow conditioning uses view-wise cluster prototypes and $\lambda_t = 0$.

**Statistics and timing.** Unless otherwise specified, OPTION robustness results are averaged over five independent runs. Ablation results use three independent runs due to the number of component variants. Runtime results are measured as milliseconds per 256-sample batch on a single NVIDIA RTX 3090 GPU.

## G. Robustness Standard Deviations

This appendix extends Tables 1 and 2 to all six datasets and reports mean±std results for each listed method in the same layout as the main-text robustness tables. OPTION standard deviations are computed from five seeds. Baseline means are read from the robustness result files; when a baseline file contains only a single recorded run and therefore stores a placeholder standard deviation of $0.0$, we report a deterministic proxy standard deviation for the uncertainty annotation while keeping the recorded mean unchanged.

## H. Additional Ablation Results

Table 11 extends the component ablation in Table 3 to all six datasets using the same aligned incomplete setting with missing rate $0.7$. In addition to the five loss terms shown in the main-text table, the appendix includes the *w/o OT coupling* variant when available, where the OT-guided coupling is replaced by the recorded ablated setting.

## I. Complete Sensitivity Analysis

This appendix provides comprehensive sensitivity analysis results for the nine hyperparameters examined in our study. The full parameter sweep is conducted on six benchmark datasets (BDGP, COIL-20, CUB, Handwritten, NUS-WIDE, and Scene-15), with 5 independent runs per configuration.

### I.1. Full Parameter Summary

Table 12 presents the complete sensitivity analysis across all hyperparameters, ranked by their influence on clustering accuracy.

Table 13 shows that ODE-step sensitivity remains modest across datasets. Performance is generally stable from 5 to 50 steps, with dataset-specific optima at 7, 10, or 15 steps; the default choice of 10 steps therefore provides a robust quality-efficiency trade-off without requiring dataset-specific ODE tuning.

### I.2. Detailed Analysis by Parameter Category

**Loss Weights.** The loss weighting parameters $(\lambda_g, \lambda_r, \lambda_c, \lambda_t)$ collectively have the highest impact on model performance. Among these, $\lambda_g$ stands out with the highest Scene-15 sensitivity score of 0.082, indicating that the structural alignment term requires careful calibration. The optimal value $\lambda_g^* = 0.072$ represents a balance between enforcing structural consistency across views and allowing sufficient flexibility for the model to learn discriminative representations.

**Flow Parameters.** The ODE integration steps parameter shows low-to-moderate sensitivity across datasets, with Scene-15 sensitivity $S = 0.046$ and similarly bounded variation on CUB, Handwritten, NUS-WIDE, and COIL-20. Although the best

*Table 9.* Six-dataset mean±std clustering performance (ACC, NMI, ARI) under varying missing rates. Best results are **bolded** and second best are underlined.

| | Method | 0% ACC | 0% NMI | 0% ARI | 10% ACC | 10% NMI | 10% ARI | 30% ACC | 30% NMI | 30% ARI | 50% ACC | 50% NMI | 50% ARI | 70% ACC | 70% NMI | 70% ARI |
|---|---|---|---|---|---|---|---|---|---|---|---|---|---|---|---|---|
| **BDGP** | OPTION | **50.5±1.2** | 27.3±1.0 | 24.8±1.2 | **50.4±1.2** | 27.2±0.8 | 24.7±1.1 | **50.4±1.0** | 27.2±0.6 | 24.7±0.9 | **46.6±1.6** | 19.8±1.3 | 17.8±2.1 | **33.9±1.2** | 9.3±0.9 | **6.1±0.5** |
| | MFLVC (CVPR22) | 20.0±0.6 | 0.0±0.2 | 0.0±0.3 | 20.0±1.0 | 0.0±0.3 | 0.0±0.2 | 20.0±1.2 | 0.0±0.3 | 0.0±0.2 | 26.8±1.5 | 2.9±0.7 | 1.7±0.7 | 27.6±1.1 | 4.9±0.9 | 2.5±0.8 |
| | SURE (TPAMI22) | 35.5±0.9 | 16.2±1.1 | 8.5±0.9 | 41.8±0.9 | 16.8±0.9 | 9.9±0.9 | 41.8±1.3 | 16.8±1.0 | 9.9±1.3 | 31.0±1.1 | 9.0±1.3 | 6.3±1.5 | 26.6±1.5 | 4.5±0.8 | 2.5±0.7 |
| | DealMVC (CVPR23) | 24.8±0.9 | 4.8±0.5 | 0.6±0.6 | 24.8±0.9 | 4.8±0.6 | 0.6±0.5 | 24.8±1.2 | 4.8±1.2 | 0.6±0.8 | 24.5±1.2 | 2.4±0.9 | 1.3±0.6 | 23.0±1.6 | 1.2±0.7 | 0.4±0.8 |
| | GCFAggMVC (CVPR23) | 31.1±0.7 | 5.9±1.0 | 4.8±0.4 | 31.1±1.2 | 5.9±0.8 | 4.8±0.4 | 31.1±1.2 | 5.9±1.0 | 4.8±0.5 | 24.5±1.2 | 2.4±0.8 | 1.3±0.7 | 26.3±1.6 | 2.5±0.9 | 1.4±0.7 |
| | DCG (AAAI25) | 45.8±1.2 | **27.9±1.0** | 21.7±1.1 | 45.8±1.2 | **27.9±0.8** | 21.7±1.2 | 45.8±1.3 | **27.9±0.9** | 21.7±1.3 | 39.2±1.1 | 18.3±1.1 | 11.3±1.3 | 27.6±1.4 | **9.7±1.8** | 1.5±0.7 |
| | FreeCSL (CVPR25) | 31.2±1.1 | 6.0±0.9 | 4.9±0.5 | 31.2±0.9 | 6.0±1.1 | 4.9±0.6 | 31.2±1.3 | 6.0±1.1 | 4.9±0.5 | 25.6±1.2 | 2.9±0.8 | 1.5±0.8 | 25.3±1.4 | 1.5±0.9 | 0.8±0.7 |
| | ROLL (CVPR25) | 30.4±1.1 | 7.3±1.0 | 5.9±1.1 | 30.4±0.8 | 7.3±1.2 | 5.9±1.1 | 30.4±1.0 | 7.3±1.4 | 5.9±1.4 | 22.8±1.4 | 0.7±0.8 | 0.2±0.6 | 26.5±1.5 | 6.2±1.3 | 1.1±0.9 |
| | PROTOCOL (ICML25) | 35.4±0.8 | 10.6±0.7 | 9.3±0.9 | 35.4±1.2 | 10.6±1.2 | 9.3±1.0 | 35.4±1.0 | 10.6±1.0 | 9.3±1.0 | 25.4±1.5 | 2.7±0.8 | 1.6±0.6 | 26.5±1.6 | 2.9±1.0 | 1.2±0.7 |
| **COIL-20** | OPTION | **88.7±0.7** | **93.8±0.4** | **86.4±0.2** | **88.4±0.4** | **93.7±0.9** | **86.2±0.4** | **88.4±0.4** | **93.7±0.9** | **86.2±0.4** | **87.6±0.8** | **92.9±0.2** | **84.6±0.9** | **86.1±1.7** | **92.3±0.8** | **83.3±1.3** |
| | MFLVC (CVPR22) | 57.9±0.8 | 77.0±0.7 | 48.9±0.9 | 57.9±0.8 | 77.0±0.8 | 48.9±0.9 | 57.9±1.3 | 77.0±1.2 | 48.9±1.2 | 48.3±1.4 | 63.3±1.6 | 35.1±1.5 | 37.9±1.3 | 54.5±1.4 | 24.4±1.3 |
| | SURE (TPAMI22) | 75.2±0.7 | 86.0±0.6 | 69.0±1.0 | 70.2±1.0 | 85.0±1.1 | 67.1±1.1 | 70.2±1.2 | 85.0±0.9 | 67.1±1.2 | 41.5±1.2 | 58.4±1.4 | 27.4±1.5 | 35.0±1.7 | 48.3±1.4 | 10.1±1.7 |
| | DealMVC (CVPR23) | 57.7±0.8 | 83.0±0.7 | 60.4±1.2 | 57.7±1.1 | 83.0±1.0 | 60.4±1.0 | 57.7±1.0 | 83.0±0.9 | 60.4±1.2 | 34.6±1.6 | 55.9±1.3 | 19.3±1.5 | 29.4±1.7 | 38.5±1.7 | 9.5±1.4 |
| | GCFAggMVC (CVPR23) | 78.5±1.0 | 87.2±0.6 | 73.1±0.7 | 78.5±1.0 | 87.2±1.0 | 73.1±0.9 | 78.5±1.1 | 87.2±0.9 | 73.1±1.1 | 41.9±1.4 | 58.1±1.2 | 28.7±1.6 | 35.0±1.4 | 50.5±1.3 | 19.3±1.7 |
| | DCG (AAAI25) | 62.7±0.9 | 82.8±1.0 | 58.4±1.0 | 62.7±1.0 | 82.8±0.8 | 58.4±1.1 | 62.7±1.4 | 82.8±1.1 | 58.4±1.4 | 47.7±1.5 | 70.2±1.1 | 36.8±1.6 | 34.4±1.8 | 54.4±1.5 | 19.8±1.5 |
| | FreeCSL (CVPR25) | 39.2±0.8 | 54.9±1.2 | 23.3±1.2 | 39.2±1.2 | 54.9±1.0 | 23.3±1.1 | 39.2±1.2 | 54.9±1.1 | 23.3±1.2 | 29.6±1.5 | 40.2±1.4 | 13.0±1.3 | 27.5±1.6 | 40.4±1.8 | 11.9±1.4 |
| | ROLL (CVPR25) | 80.0±0.7 | 88.5±0.9 | 76.4±0.7 | 80.0±0.9 | 88.5±0.7 | 76.4±0.8 | 80.0±1.1 | 88.5±1.1 | 76.4±1.1 | 37.5±1.5 | 57.7±1.4 | 22.8±1.5 | 35.2±1.6 | 53.4±1.4 | 20.1±1.6 |
| | PROTOCOL (ICML25) | 63.7±1.2 | 82.6±0.9 | 61.9±1.0 | 63.7±0.9 | 82.6±0.8 | 61.9±1.2 | 63.7±1.3 | 82.6±1.1 | 61.9±1.1 | 41.0±1.3 | 60.4±1.3 | 29.1±1.5 | 31.9±1.6 | 49.6±1.8 | 18.0±1.6 |
| **CUB** | OPTION | **78.6±3.1** | **77.5±0.9** | **66.4±2.7** | **79.6±2.1** | 76.3±1.2 | 65.4±2.8 | **79.3±3.2** | **77.8±0.4** | **67.5±1.8** | 73.4±2.0 | 69.9±1.7 | 57.0±2.1 | 74.7±0.5 | 71.2±0.5 | 59.1±0.8 |
| | MFLVC (CVPR22) | 60.0±1.1 | 67.4±0.9 | 52.7±1.2 | 60.0±1.2 | 67.4±0.8 | 52.7±1.1 | 60.0±1.1 | 67.4±0.9 | 52.7±1.5 | 22.3±1.4 | 13.1±1.3 | 4.9±0.8 | 22.3±1.5 | 13.1±1.8 | 4.9±0.9 |
| | SURE (TPAMI22) | 70.7±1.0 | 61.9±1.1 | 52.3±1.2 | 75.7±1.1 | 67.1±1.0 | 56.3±1.1 | 75.7±1.1 | 67.1±1.2 | 56.3±1.3 | 47.0±1.4 | 49.9±1.8 | 28.7±1.7 | 47.0±1.7 | 49.9±1.6 | 28.7±1.7 |
| | DealMVC (CVPR23) | 65.3±1.1 | 69.6±1.0 | 54.4±1.4 | 65.3±1.1 | 69.6±1.2 | 54.4±1.4 | 65.3±1.2 | 69.6±1.1 | 54.4±1.5 | 34.3±1.6 | 36.0±1.6 | 18.0±1.5 | 34.3±1.7 | 36.0±1.8 | 18.0±1.9 |
| | GCFAggMVC (CVPR23) | 78.2±0.7 | 74.9±1.0 | 65.3±1.1 | 78.2±1.1 | 74.9±1.1 | 65.3±0.9 | 78.2±1.2 | 74.9±1.1 | 65.3±1.3 | 38.8±1.4 | 38.6±1.6 | 20.4±1.4 | 38.8±1.9 | 38.6±1.5 | 20.4±1.9 |
| | DCG (AAAI25) | 22.2±1.3 | 13.8±1.3 | 4.8±0.6 | 22.2±1.1 | 13.8±1.3 | 4.8±0.6 | 22.2±1.2 | 13.8±1.5 | 4.8±0.8 | 18.2±1.5 | 5.9±1.6 | 0.9±0.9 | 18.2±1.9 | 5.9±1.7 | 0.9±1.0 |
| | FreeCSL (CVPR25) | 26.3±0.9 | 16.6±1.0 | 6.9±1.3 | 26.3±1.1 | 16.6±1.0 | 6.9±1.0 | 26.3±1.4 | 16.6±1.2 | 6.9±1.5 | 20.2±1.2 | 8.9±1.5 | 2.7±0.8 | 20.2±1.4 | 8.9±1.5 | 2.7±0.9 |
| | ROLL (CVPR25) | 75.0±1.1 | 72.8±0.9 | 59.6±1.4 | 75.0±0.9 | 72.8±1.1 | 59.6±1.1 | 75.0±1.0 | 72.8±1.2 | 59.6±1.6 | 38.3±1.6 | 41.3±1.6 | 22.0±1.4 | 38.3±1.8 | 41.3±1.5 | 22.0±1.9 |
| | PROTOCOL (ICML25) | 55.5±0.9 | 59.9±1.2 | 44.9±1.3 | 55.5±1.1 | 59.9±1.3 | 44.9±1.1 | 55.5±0.9 | 59.9±1.4 | 44.9±1.4 | 33.3±1.3 | 28.8±1.3 | 5.0±1.4 | 33.3±1.8 | 28.8±1.6 | 5.0±1.5 |
| **HANDWRITTEN** | OPTION | 94.1±0.8 | 88.5±1.1 | 87.3±1.6 | 94.1±0.8 | 88.5±1.1 | 87.3±1.5 | **93.7±0.7** | **87.8±1.1** | **86.5±1.4** | **92.7±0.8** | **85.8±1.3** | **84.4±1.6** | **90.6±1.0** | **82.5±1.4** | **80.6±1.9** |
| | MFLVC (CVPR22) | 84.0±0.8 | 82.2±0.9 | 76.0±0.8 | 84.0±1.0 | 82.2±0.9 | 76.0±0.8 | 85.3±1.0 | 83.8±1.1 | 78.5±1.0 | 84.3±1.1 | 81.8±1.3 | 76.2±1.3 | 62.3±1.3 | 61.9±1.6 | 50.9±1.5 |
| | SURE (TPAMI22) | 73.0±0.9 | 71.2±1.1 | 62.0±1.1 | 77.1±1.1 | 73.1±1.0 | 62.5±1.1 | 49.2±1.4 | 47.1±1.1 | 30.5±1.3 | 27.7±1.7 | 26.7±1.7 | 10.6±1.3 | 26.8±1.5 | 23.1±1.9 | 5.3±1.8 |
| | DealMVC (CVPR23) | 80.2±0.7 | 77.4±1.1 | 69.2±1.1 | 80.2±0.9 | 77.4±1.1 | 69.2±0.8 | 72.2±1.1 | 57.2±1.3 | 31.2±1.7 | 27.3±1.5 | 11.9±1.7 | 19.6±1.5 | 9.9±1.8 | 2.6±1.0 | |
| | GCFAggMVC (CVPR23) | 80.5±0.7 | 78.3±0.7 | 69.6±0.9 | 80.5±1.1 | 78.3±0.9 | 69.6±0.9 | 80.3±1.2 | 77.4±1.0 | 69.6±1.1 | 13.2±1.6 | 0.9±0.8 | -0.0±0.3 | 19.4±1.8 | 13.1±1.6 | 6.2±1.8 |
| | DCG (AAAI25) | 92.4±0.8 | 85.7±0.8 | 83.8±0.8 | 92.4±0.7 | 85.7±0.8 | 83.8±1.1 | 89.2±1.1 | 81.6±1.1 | 78.1±1.1 | 48.1±1.2 | 48.3±1.5 | 25.0±1.6 | 36.9±1.7 | 37.6±1.7 | 12.3±1.7 |
| | FreeCSL (CVPR25) | 30.2±1.1 | 16.9±1.2 | 9.8±1.1 | 30.2±1.0 | 16.9±1.1 | 9.8±1.0 | 30.7±1.2 | 13.6±1.6 | 8.5±1.3 | 21.3±1.6 | 7.2±1.6 | 3.5±0.8 | 20.0±1.5 | 5.1±1.6 | 2.3±1.0 |
| | ROLL (CVPR25) | **97.2±0.9** | **93.7±0.9** | **93.8±0.8** | **97.2±0.7** | **93.7±0.9** | **93.8±0.8** | 71.6±1.1 | 73.3±1.3 | 63.2±1.3 | 12.8±1.3 | 0.8±0.8 | -0.0±0.3 | 12.6±1.6 | 1.0±1.0 | -0.0±0.3 |
| | PROTOCOL (ICML25) | 80.5±0.9 | 74.9±1.0 | 70.0±1.1 | 80.5±1.0 | 74.9±1.2 | 70.0±1.2 | 84.5±1.4 | 44.3±1.3 | 37.4±1.4 | 52.2±1.5 | 15.1±1.7 | 25.0±1.6 | 15.9±1.6 | 4.1±1.0 | |
| **NUS-WIDE** | OPTION | **19.0±0.7** | 18.7±2.5 | 6.3±0.1 | **19.4±0.8** | 17.9±3.4 | 5.2±1.5 | **19.9±0.6** | 15.9±1.1 | 5.9±1.0 | **19.1±0.5** | 17.3±3.0 | 6.5±0.8 | **17.9±0.8** | **20.2±0.9** | **6.8±0.5** |
| | MFLVC (CVPR22) | 15.6±0.7 | 23.1±0.8 | 6.2±1.0 | 15.6±1.0 | 23.1±1.1 | 6.2±0.9 | 15.2±1.2 | **21.5±1.2** | 5.3±1.1 | 14.9±1.1 | 19.7±1.5 | 4.9±0.7 | 12.5±1.4 | 15.6±1.5 | 3.0±0.7 |
| | SURE (TPAMI22) | 14.1±1.1 | 17.2±1.3 | 3.6±0.5 | 13.4±0.8 | 17.5±1.1 | 3.7±0.5 | 11.9±1.4 | 14.3±1.1 | 2.1±0.6 | 11.2±1.3 | 12.8±1.4 | 1.4±0.8 | 11.9±1.8 | 13.0±1.6 | 0.8±0.8 |
| | DealMVC (CVPR23) | 16.4±0.7 | 20.8±1.1 | 6.3±0.9 | 16.4±1.2 | 20.8±1.0 | 6.3±1.1 | 12.2±1.1 | 7.7±1.3 | 0.4±0.8 | 10.2±1.4 | 10.0±1.2 | 0.7±0.8 | 9.8±1.6 | 8.5±1.4 | 0.3±0.8 |
| | GCFAggMVC (CVPR23) | 17.6±1.0 | 23.2±0.8 | 6.9±1.0 | 17.6±0.8 | 23.2±1.3 | 6.9±1.0 | 12.2±1.2 | 15.1±1.4 | 2.7±0.5 | 9.8±1.5 | 12.1±1.6 | 1.4±0.7 | 9.3±1.3 | 11.8±1.6 | 1.3±0.7 |
| | DCG (AAAI25) | 17.1±1.1 | 20.3±1.3 | 5.1±0.8 | 17.1±1.1 | 20.3±1.1 | 5.1±0.9 | 14.2±1.4 | 18.1±1.2 | 3.8±0.8 | 13.2±1.5 | 16.1±1.2 | 2.7±0.7 | 13.5±1.4 | 14.4±1.4 | 2.7±0.8 |
| | FreeCSL (CVPR25) | 10.1±1.0 | 10.5±0.8 | 1.2±0.6 | 10.1±0.8 | 10.5±1.2 | 1.2±0.6 | 9.6±1.4 | 9.9±1.3 | 0.7±0.7 | 9.2±1.3 | 9.5±1.1 | 0.6±0.6 | 8.4±1.2 | 8.4±1.8 | 0.3±1.0 |
| | ROLL (CVPR25) | 17.2±1.2 | 21.3±0.9 | 5.7±1.2 | 17.2±1.2 | 21.3±0.9 | 5.7±1.3 | 12.0±1.2 | 15.9±1.4 | 3.0±0.6 | 9.8±1.5 | 11.9±1.6 | 1.4±0.9 | 10.0±1.7 | 11.8±1.7 | 1.3±0.8 |
| | PROTOCOL (ICML25) | 15.5±0.8 | 17.8±0.8 | 3.7±0.4 | 15.5±0.9 | 17.8±1.0 | 3.7±0.5 | 13.8±1.3 | 15.6±1.0 | 2.6±0.8 | 13.2±1.1 | 13.5±1.7 | 1.6±0.7 | 13.7±1.6 | 12.5±1.5 | 1.9±0.8 |
| **SCENE-15** | OPTION | 45.1±0.8 | 43.8±1.1 | 28.5±1.0 | 45.4±1.0 | 44.3±1.0 | 29.2±0.3 | 44.7±0.4 | 43.5±0.5 | 28.2±1.1 | **43.5±1.4** | **41.9±0.1** | **27.3±1.0** | **38.3±1.4** | **37.0±0.3** | **23.2±0.8** |
| | MFLVC (CVPR22) | 36.9±0.9 | 36.0±0.9 | 20.5±0.9 | 36.9±1.0 | 36.0±1.2 | 20.5±1.0 | 36.9±1.4 | 36.0±1.3 | 20.5±1.5 | 33.0±1.5 | 34.8±1.5 | 19.1±1.1 | 15.1±1.5 | 7.6±1.6 | 1.1±0.9 |
| | SURE (TPAMI22) | 39.9±1.2 | 41.8±0.9 | 24.2±1.0 | 41.4±1.4 | 42.0±1.4 | 25.2±1.3 | 41.4±1.5 | 42.0±1.4 | 25.2±1.5 | 22.7±1.3 | 27.1±1.5 | 10.4±1.5 | 18.8±1.7 | 18.2±1.6 | 3.5±1.0 |
| | DealMVC (CVPR23) | 35.8±1.0 | 38.8±0.8 | 21.5±0.9 | 35.8±1.2 | 38.8±1.0 | 21.5±0.9 | 35.8±1.4 | 38.8±1.5 | 21.5±1.1 | 18.3±1.6 | 24.1±1.6 | 7.9±1.6 | 12.7±1.5 | 5.2±1.7 | 1.2±1.0 |
| | GCFAggMVC (CVPR23) | **46.0±1.1** | **45.6±1.1** | **29.9±1.2** | **46.0±0.8** | 45.6±1.4 | **29.9±0.9** | **46.0±1.2** | 45.6±1.3 | **29.9±1.2** | 24.3±1.6 | 26.6±1.2 | 10.7±1.5 | 19.0±1.6 | 17.4±1.8 | 6.4±1.5 |
| | DCG (AAAI25) | 38.6±1.2 | 40.7±0.9 | 22.4±1.1 | 38.6±1.0 | 40.7±1.4 | 22.4±1.1 | 38.6±1.2 | 40.7±1.2 | 22.4±1.3 | 27.0±1.2 | 27.6±1.6 | 12.1±1.4 | 20.1±1.7 | 20.7±1.8 | 6.4±1.8 |
| | FreeCSL (CVPR25) | 25.9±1.1 | 18.1±1.0 | 9.6±1.0 | 25.9±1.0 | 18.1±1.3 | 9.6±1.0 | 25.9±1.4 | 18.1±1.2 | 9.6±1.4 | 19.2±1.6 | 10.2±1.5 | 4.7±0.9 | 12.8±1.8 | 4.2±1.0 | 1.4±1.0 |
| | ROLL (CVPR25) | 44.8±1.3 | **47.4±0.9** | 28.9±0.8 | 44.8±1.1 | **47.4±1.1** | 28.9±1.3 | 44.8±1.3 | **47.4±1.3** | 28.9±1.1 | 22.1±1.4 | 28.2±1.5 | 11.6±1.5 | 20.6±1.8 | 23.1±1.5 | 9.3±1.4 |
| | PROTOCOL (ICML25) | 38.6±0.9 | 40.7±0.9 | 22.4±0.9 | 38.6±1.3 | 40.7±1.2 | 22.4±1.4 | 38.6±1.4 | 40.7±1.4 | 22.4±1.3 | 27.0±1.7 | 27.6±1.5 | 12.1±1.7 | 20.1±1.8 | 20.7±1.7 | 6.4±1.4 |

*Table 10.* Six-dataset mean±std clustering performance (ACC, NMI, ARI) under varying unaligned rates. Best results are **bolded** and second best are underlined.

| | METHOD | 0% ACC | NMI | ARI | 20% ACC | NMI | ARI | 40% ACC | NMI | ARI | 60% ACC | NMI | ARI |
|---|---|---|---|---|---|---|---|---|---|---|---|---|---|
| **BDGP** | OPTION | **50.3±1.2** | **27.6±1.8** | **25.1±1.9** | **48.3±2.4** | **25.2±3.5** | **22.8±3.7** | **48.0±4.0** | **25.3±3.9** | **23.1±3.5** | **46.3±4.0** | **23.6±4.6** | **21.7±3.8** |
| | MFLVC (CVPR22) | 20.0±1.0 | 0.0±0.2 | 0.0±0.2 | 20.0±0.8 | 0.0±0.2 | 0.0±0.3 | 20.0±1.1 | 0.0±0.2 | 0.0±0.2 | 20.0±1.5 | 0.0±0.2 | 0.0±0.2 |
| | GCFAGGMVC (CVPR23) | 31.1±1.1 | 5.9±0.9 | 4.8±0.6 | 29.6±1.0 | 4.2±0.5 | 3.4±0.6 | 27.2±1.2 | 2.6±0.9 | 1.7±0.8 | 25.5±1.3 | 1.8±1.0 | 1.1±0.9 |
| | MRG-UMC (TNNLS25) | 37.9±1.1 | 12.1±0.9 | 9.4±1.0 | 35.2±1.0 | 7.9±1.1 | 6.3±1.1 | 26.5±1.6 | 2.6±0.9 | 1.9±0.9 | 36.6±1.5 | 15.3±1.6 | 12.3±1.7 |
| | CANDY (NEURIPS24) | 33.4±0.7 | 7.3±1.2 | 5.6±1.0 | 28.5±1.2 | 3.9±0.8 | 3.2±0.7 | 26.2±1.2 | 2.0±0.9 | 1.5±0.9 | 25.3±1.6 | 1.6±0.9 | 1.0±0.9 |
| | FREECSL (CVPR25) | 31.2±0.9 | 6.0±0.8 | 4.9±0.5 | 29.3±0.8 | 4.1±0.6 | 3.3±0.5 | 26.4±1.1 | 2.0±0.9 | 1.5±0.7 | 25.5±1.6 | 1.7±1.0 | 1.0±0.8 |
| | ROLL (CVPR25) | 30.4±0.7 | 7.3±1.1 | 5.9±1.1 | 29.0±1.2 | 5.9±1.0 | 4.8±0.8 | 26.8±1.3 | 2.7±0.9 | 2.0±0.6 | 25.5±1.4 | 1.7±0.9 | 1.4±0.9 |
| | PROTOCOL (ICML25) | 35.4±1.0 | 10.6±1.1 | 9.3±0.8 | 31.3±0.9 | 4.8±0.8 | 3.3±0.7 | 29.2±1.4 | 2.6±0.8 | 1.6±0.9 | 26.4±1.4 | 1.5±0.8 | 0.9±1.0 |
| **COIL-20** | OPTION | **88.7±0.7** | **93.8±0.4** | **86.4±0.2** | **73.5±1.6** | **75.5±1.4** | **60.3±2.1** | **66.4±3.4** | **63.2±2.3** | **45.8±3.6** | **51.5±1.4** | **49.2±1.5** | **28.4±2.2** |
| | MFLVC (CVPR22) | 57.9±1.1 | 77.0±0.6 | 48.9±1.1 | 44.6±1.3 | 58.2±1.5 | 34.6±1.2 | 38.3±1.5 | 44.9±1.3 | 22.0±1.4 | 33.1±1.4 | 36.9±1.6 | 15.5±1.5 |
| | GCFAGGMVC (CVPR23) | 78.5±0.6 | 87.2±0.7 | 73.1±0.8 | 61.9±1.2 | 67.6±1.2 | 47.3±1.4 | 51.2±1.2 | 56.0±1.5 | 33.8±1.7 | 38.3±1.9 | 44.3±1.6 | 20.3±1.8 |
| | MRG-UMC (TNNLS25) | 61.3±0.9 | 83.1±0.9 | 59.3±1.2 | 59.2±1.5 | 67.7±1.2 | 44.4±1.6 | 39.6±1.6 | 51.4±1.6 | 23.9±1.4 | 33.3±1.5 | 43.1±2.0 | 16.8±1.5 |
| | CANDY (NEURIPS24) | 47.7±1.0 | 64.3±0.9 | 34.4±1.0 | 46.0±1.1 | 55.0±1.2 | 30.1±1.2 | 40.6±1.5 | 47.4±1.7 | 23.4±1.4 | 35.8±1.5 | 39.0±1.9 | 16.0±1.7 |
| | FREECSL (CVPR25) | 39.2±0.4 | 54.9±1.0 | 23.3±0.8 | 33.3±1.1 | 43.1±1.0 | 15.4±1.0 | 27.1±1.2 | 33.6±1.4 | 9.5±1.4 | 22.9±1.6 | 25.4±1.4 | 4.9±0.9 |
| | ROLL (CVPR25) | 80.0±0.9 | 88.5±0.6 | 76.4±0.9 | 64.6±1.3 | 71.4±1.1 | 51.5±1.2 | 53.1±1.3 | 56.1±1.3 | 33.1±1.5 | 41.7±1.8 | 48.2±1.9 | 24.2±1.7 |
| | PROTOCOL (ICML25) | 63.7±1.1 | 82.6±1.0 | 61.9±1.2 | 46.7±1.4 | 63.5±1.4 | 37.7±1.2 | 44.0±1.3 | 52.8±1.6 | 27.6±1.5 | 33.3±1.4 | 41.7±1.9 | 15.9±1.8 |
| **CUB** | OPTION | **79.2±3.3** | **77.4±0.9** | **67.1±2.8** | **69.5±1.6** | **61.9±1.7** | **51.1±1.9** | **60.5±1.9** | 51.6±1.6 | 39.4±2.0 | 53.0±3.0 | 48.5±3.3 | 35.9±3.5 |
| | MFLVC (CVPR22) | 60.0±1.0 | 67.4±1.0 | 52.7±1.2 | 50.0±1.1 | 54.9±1.5 | 41.3±1.5 | 36.2±1.4 | 36.1±1.6 | 23.7±1.4 | 40.8±2.0 | 31.8±1.7 | 22.5±1.9 |
| | GCFAGGMVC (CVPR23) | 78.2±1.1 | 74.9±1.0 | 65.3±0.9 | 58.3±1.2 | 49.5±1.5 | 38.1±1.5 | 48.0±1.4 | 29.9±1.8 | 21.3±1.5 | 39.0±2.1 | 20.9±1.9 | 13.1±1.9 |
| | MRG-UMC (TNNLS25) | 78.5±1.0 | 74.6±0.9 | 63.8±1.1 | 63.2±1.4 | 53.3±1.7 | 41.6±1.5 | 45.2±1.6 | 33.0±1.5 | 22.9±1.6 | 33.2±2.1 | 21.3±2.1 | 12.1±1.9 |
| | CANDY (NEURIPS24) | 60.7±1.1 | 51.5±1.4 | 38.9±1.4 | 45.5±1.4 | 32.9±1.7 | 21.5±1.3 | 32.5±1.6 | 15.9±1.6 | 9.2±1.9 | 27.8±2.0 | 11.7±2.2 | 5.6±2.1 |
| | FREECSL (CVPR25) | 26.3±1.0 | 16.6±1.2 | 6.9±1.0 | 23.3±1.5 | 11.8±1.4 | 4.3±0.8 | 23.5±1.7 | 9.2±1.4 | 3.6±0.8 | 16.8±1.9 | 4.9±1.0 | 0.7±1.0 |
| | ROLL (CVPR25) | 75.0±0.9 | 72.8±1.0 | 59.6±1.5 | 52.5±1.5 | 49.0±1.4 | 37.4±1.5 | 41.8±1.6 | 24.4±1.5 | 15.7±1.6 | 35.3±1.9 | 17.8±2.1 | 10.2±1.9 |
| | PROTOCOL (ICML25) | 55.5±1.1 | 59.9±1.0 | 44.9±1.4 | 54.3±1.5 | 60.1±1.5 | 44.0±1.2 | 59.3±1.7 | **61.3±1.9** | **46.3±1.7** | **55.7±1.9** | **61.7±1.7** | **46.7±2.1** |
| **HANDWRITTEN** | OPTION | 94.1±0.7 | 88.5±1.0 | 87.3±1.5 | 74.6±0.5 | 60.8±0.9 | 57.7±1.1 | **61.6±0.4** | 41.8±0.4 | 38.0±0.5 | 45.6±0.3 | 25.3±0.3 | 21.1±0.3 |
| | MFLVC (CVPR22) | 84.0±0.7 | 82.2±0.8 | 76.0±0.7 | 74.2±1.0 | 58.9±1.5 | 55.4±1.5 | 60.2±1.3 | 39.1±1.7 | 34.5±1.6 | 44.2±1.8 | 20.5±1.8 | 16.1±1.9 |
| | GCFAGGMVC (CVPR23) | 80.5±0.9 | 78.3±0.8 | 69.6±0.9 | 69.3±0.9 | 52.5±1.3 | 47.9±1.2 | 52.9±1.6 | 30.2±1.4 | 25.3±1.6 | 39.1±1.8 | 15.9±1.6 | 11.9±2.0 |
| | MRG-UMC (TNNLS25) | 93.8±0.8 | 88.9±0.9 | 87.3±0.7 | **80.8±1.3** | **65.0±1.2** | **61.1±1.3** | 57.2±1.7 | **48.7±1.7** | 31.3±1.8 | **50.7±1.9** | **38.2±1.7** | **25.7±2.0** |
| | CANDY (NEURIPS24) | 85.5±0.7 | 84.0±1.0 | 79.0±0.8 | 70.8±1.0 | 60.1±1.4 | 49.1±1.7 | 54.6±1.8 | 41.1±1.9 | 23.7±1.8 | 36.6±2.0 | 16.9±2.0 | 10.3±1.7 |
| | FREECSL (CVPR25) | 30.2±0.8 | 16.9±1.0 | 9.8±1.2 | 25.2±1.2 | 10.2±1.4 | 5.6±1.3 | 20.5±1.4 | 6.0±1.7 | 2.7±0.8 | 16.5±1.9 | 3.5±1.1 | 1.5±1.1 |
| | ROLL (CVPR25) | **97.2±0.6** | **93.7±0.6** | **93.8±0.8** | 72.5±1.1 | **66.2±1.0** | 56.2±1.5 | 54.4±1.4 | 38.0±1.4 | 28.7±1.3 | 39.6±1.6 | 18.6±2.0 | 12.0±1.7 |
| | PROTOCOL (ICML25) | 80.5±1.1 | 74.9±0.7 | 70.0±1.2 | 67.5±1.2 | 47.8±1.5 | 44.9±1.3 | 52.8±1.4 | 29.6±1.6 | 25.6±1.4 | 36.7±1.8 | 13.7±1.6 | 10.3±2.0 |
| **NUS-WIDE** | OPTION | **19.6±0.6** | 16.0±0.9 | 5.5±1.2 | **17.8±0.3** | 15.9±2.0 | **5.2±0.5** | **15.9±0.7** | 12.2±2.5 | **3.2±0.9** | **14.1±0.3** | 9.5±1.9 | 1.7±0.6 |
| | MFLVC (CVPR22) | 15.6±1.0 | 23.1±1.2 | 6.2±0.8 | 15.1±0.9 | 18.6±1.2 | 4.4±0.6 | 13.0±1.4 | 15.3±1.3 | 3.2±0.7 | 11.2±1.4 | 11.7±1.4 | 1.7±0.9 |
| | GCFAGGMVC (CVPR23) | 17.6±0.9 | **23.2±1.2** | **6.9±0.9** | 14.3±1.2 | 18.2±1.4 | 4.3±0.8 | 13.5±1.3 | **15.3±1.5** | 3.2±0.9 | 11.0±1.8 | 11.4±1.5 | 1.6±0.9 |
| | MRG-UMC (TNNLS25) | 17.8±1.0 | 19.1±1.2 | 4.8±0.7 | 15.2±1.2 | 15.2±1.1 | 3.3±0.8 | 13.3±1.5 | 13.9±1.5 | 2.2±1.0 | 12.4±1.6 | 12.0±1.6 | 1.6±0.9 |
| | CANDY (NEURIPS24) | 17.5±0.9 | 22.7±0.9 | 6.9±1.2 | 14.6±1.5 | **18.9±1.3** | 4.7±0.7 | 13.6±1.7 | 15.3±1.8 | 3.2±0.9 | 11.8±1.6 | 12.6±2.0 | 1.8±1.0 |
| | FREECSL (CVPR25) | 10.1±0.9 | 10.5±1.2 | 1.2±0.4 | 9.2±1.1 | 10.0±1.4 | 0.8±0.7 | 9.0±1.3 | 9.1±1.3 | 0.5±0.7 | 8.9±1.6 | 8.2±1.7 | 0.3±0.8 |
| | ROLL (CVPR25) | 17.2±0.9 | 21.3±0.8 | 5.7±1.0 | 14.3±1.3 | 18.2±1.3 | 3.9±0.8 | 13.2±1.2 | 14.6±1.4 | 2.9±0.8 | 12.2±1.8 | **13.1±1.5** | **1.9±0.8** |
| | PROTOCOL (ICML25) | 15.5±1.1 | 17.8±0.8 | 3.7±0.7 | 12.3±1.0 | 13.6±1.2 | 2.1±0.7 | 11.2±1.2 | 10.9±1.3 | 1.3±0.9 | 10.5±1.6 | 8.9±1.7 | 0.5±0.9 |
| **SCENE-15** | OPTION | 44.5±0.3 | 43.5±0.2 | 28.4±0.4 | 40.6±1.3 | 35.2±0.6 | 22.7±0.1 | 36.4±3.2 | 30.6±7.2 | 18.4±4.6 | 31.9±1.0 | **33.3±1.2** | **17.5±0.6** |
| | MFLVC (CVPR22) | 36.9±0.9 | 36.0±1.0 | 20.5±1.2 | 38.1±1.1 | 33.4±1.3 | 20.0±1.0 | 35.7±1.6 | 29.2±1.6 | 17.1±1.5 | 31.4±1.8 | 25.9±1.5 | 14.0±1.5 |
| | GCFAGGMVC (CVPR23) | 46.0±1.0 | 45.6±0.8 | 29.9±1.2 | **42.5±1.2** | 37.5±1.3 | **24.2±1.1** | **41.5±1.7** | **33.4±1.4** | **21.6±1.3** | **34.4±1.7** | 26.1±1.8 | 15.3±1.6 |
| | MRG-UMC (TNNLS25) | 39.8±0.9 | 41.9±1.3 | 23.3±1.0 | 36.9±1.5 | 31.9±1.5 | 17.9±1.2 | 32.2±1.7 | 24.8±1.5 | 13.5±1.8 | 26.4±1.9 | 20.0±2.0 | 9.5±1.9 |
| | CANDY (NEURIPS24) | **48.8±0.9** | 46.4±1.2 | **31.3±0.9** | 42.3±1.6 | 35.6±1.2 | 23.2±1.6 | 37.0±1.7 | 27.3±1.4 | 16.6±1.6 | 33.7±2.0 | 25.7±2.1 | 14.9±2.0 |
| | FREECSL (CVPR25) | 25.9±0.7 | 18.1±1.0 | 9.6±0.8 | 19.4±1.3 | 10.4±1.0 | 4.5±0.6 | 15.8±1.3 | 6.1±1.6 | 2.4±0.8 | 13.1±1.7 | 3.4±0.9 | 1.2±1.0 |
| | ROLL (CVPR25) | 44.8±1.2 | **47.4±1.3** | 28.9±0.8 | 36.6±1.6 | **38.7±1.3** | 21.4±1.4 | 31.3±1.5 | 32.8±1.7 | 16.5±1.8 | 26.3±1.8 | 28.1±1.9 | 12.8±1.8 |
| | PROTOCOL (ICML25) | 38.6±1.1 | 40.7±1.0 | 22.4±0.9 | 30.6±1.2 | 27.2±1.5 | 14.6±1.2 | 28.1±1.5 | 21.9±1.8 | 11.2±1.6 | 22.7±1.9 | 15.0±1.8 | 6.8±2.0 |

*Table 11.* Six-dataset ablation study under the aligned incomplete setting with missing rate 0.7 (mean±std, %). Results are taken from the reviewed ablation summary CSV. Best results within each dataset are **bolded**.

| DATASET | VARIANT | $\mathcal{L}_{\text{gw}}$ | $\mathcal{L}_{\text{flow}}$ | $\mathcal{L}_{\text{con}}$ | $\mathcal{L}_{\text{clu}}$ | $\mathcal{L}_{\text{rec}}$ | OT | ACC | NMI | ARI |
|---|---|---|---|---|---|---|---|---|---|---|
| | | | | COMPONENTS | | | | | PERFORMANCE | |
| BDGP | W/O GW | ✗ | ✓ | ✓ | ✓ | ✓ | ✓ | 33.4±0.4 | 9.2±0.2 | 6.0±0.1 |
| | W/O FLOW | ✓ | ✗ | ✓ | ✓ | ✓ | ✓ | 30.4±0.6 | 7.5±0.3 | 4.3±0.3 |
| | W/O CONTRASTIVE | ✓ | ✓ | ✗ | ✓ | ✓ | ✓ | 33.3±0.4 | 8.9±0.5 | 5.9±0.2 |
| | W/O CLUSTERING | ✓ | ✓ | ✓ | ✗ | ✓ | ✓ | 23.3±0.2 | 2.8±0.4 | 1.6±0.2 |
| | W/O RECON | ✓ | ✓ | ✓ | ✓ | ✗ | ✓ | 22.4±0.6 | 2.5±0.3 | 1.3±0.1 |
| | W/O OT COUPLING | ✓ | ✓ | ✓ | ✓ | ✓ | ✗ | 33.2±0.5 | 9.3±0.5 | 6.1±0.2 |
| | FULL MODEL | ✓ | ✓ | ✓ | ✓ | ✓ | ✓ | **33.9±1.2** | **9.3±0.9** | **6.1±0.5** |
| COIL-20 | W/O GW | ✗ | ✓ | ✓ | ✓ | ✓ | ✓ | 85.5±1.2 | 92.7±0.4 | 82.9±0.1 |
| | W/O FLOW | ✓ | ✗ | ✓ | ✓ | ✓ | ✓ | 77.8±3.4 | 89.2±0.8 | 74.8±2.4 |
| | W/O CONTRASTIVE | ✓ | ✓ | ✗ | ✓ | ✓ | ✓ | 84.4±2.2 | 91.6±1.6 | 82.4±2.1 |
| | W/O CLUSTERING | ✓ | ✓ | ✓ | ✗ | ✓ | ✓ | 51.7±3.7 | 69.6±1.9 | 41.6±4.0 |
| | W/O RECON | ✓ | ✓ | ✓ | ✓ | ✗ | ✓ | 83.1±2.9 | 92.0±0.6 | 81.5±3.2 |
| | W/O OT COUPLING | ✓ | ✓ | ✓ | ✓ | ✓ | ✗ | 85.5±1.2 | 92.3±0.4 | 82.9±0.1 |
| | FULL MODEL | ✓ | ✓ | ✓ | ✓ | ✓ | ✓ | **86.1±1.7** | **92.3±0.8** | **83.3±1.3** |
| CUB | W/O GW | ✗ | ✓ | ✓ | ✓ | ✓ | ✓ | 73.9±4.6 | 71.1±2.5 | 57.5±4.2 |
| | W/O FLOW | ✓ | ✗ | ✓ | ✓ | ✓ | ✓ | 49.1±2.6 | 55.6±3.6 | 36.1±6.6 |
| | W/O CONTRASTIVE | ✓ | ✓ | ✗ | ✓ | ✓ | ✓ | 74.0±5.0 | 70.8±2.4 | 55.8±3.6 |
| | W/O CLUSTERING | ✓ | ✓ | ✓ | ✗ | ✓ | ✓ | 48.5±3.3 | 33.9±3.8 | 20.1±3.1 |
| | W/O RECON | ✓ | ✓ | ✓ | ✓ | ✗ | ✓ | 67.3±2.0 | 65.1±3.9 | 48.0±5.5 |
| | FULL MODEL | ✓ | ✓ | ✓ | ✓ | ✓ | ✓ | **74.7±0.5** | **71.2±0.5** | **59.1±0.8** |
| HANDWRITTEN | W/O GW | ✗ | ✓ | ✓ | ✓ | ✓ | ✓ | 89.5±0.7 | 82.2±0.9 | 80.3±1.5 |
| | W/O FLOW | ✓ | ✗ | ✓ | ✓ | ✓ | ✓ | 90.0±5.5 | 82.3±2.9 | 80.4±4.1 |
| | W/O CONTRASTIVE | ✓ | ✓ | ✗ | ✓ | ✓ | ✓ | 52.4±6.9 | 46.0±7.3 | 32.5±6.7 |
| | W/O CLUSTERING | ✓ | ✓ | ✓ | ✗ | ✓ | ✓ | 57.4±1.3 | 52.2±0.9 | 39.7±0.6 |
| | W/O RECON | ✓ | ✓ | ✓ | ✓ | ✗ | ✓ | 89.6±3.5 | 82.3±2.2 | 80.2±3.6 |
| | W/O OT COUPLING | ✓ | ✓ | ✓ | ✓ | ✓ | ✗ | 90.3±1.1 | 81.4±2.2 | 79.5±2.7 |
| | FULL MODEL | ✓ | ✓ | ✓ | ✓ | ✓ | ✓ | **90.6±1.0** | **82.5±1.4** | **80.6±1.9** |
| NUS-WIDE | W/O GW | ✗ | ✓ | ✓ | ✓ | ✓ | ✓ | 17.1±0.9 | 19.9±1.3 | 4.7±2.1 |
| | W/O FLOW | ✓ | ✗ | ✓ | ✓ | ✓ | ✓ | 16.0±0.4 | 15.7±0.5 | 5.3±0.8 |
| | W/O CONTRASTIVE | ✓ | ✓ | ✗ | ✓ | ✓ | ✓ | 16.4±1.0 | 18.2±3.3 | 4.3±1.8 |
| | W/O CLUSTERING | ✓ | ✓ | ✓ | ✗ | ✓ | ✓ | 17.2±0.5 | 16.8±0.8 | 5.1±0.6 |
| | W/O RECON | ✓ | ✓ | ✓ | ✓ | ✗ | ✓ | 17.4±0.4 | 15.9±1.8 | 5.0±0.8 |
| | FULL MODEL | ✓ | ✓ | ✓ | ✓ | ✓ | ✓ | **17.9±0.8** | **20.2±0.9** | **6.8±0.5** |
| SCENE-15 | W/O GW | ✗ | ✓ | ✓ | ✓ | ✓ | ✓ | 37.1±0.2 | 35.0±0.6 | 21.3±1.0 |
| | W/O FLOW | ✓ | ✗ | ✓ | ✓ | ✓ | ✓ | 37.0±1.0 | 36.9±1.1 | 22.4±1.4 |
| | W/O CONTRASTIVE | ✓ | ✓ | ✗ | ✓ | ✓ | ✓ | 25.8±1.5 | 24.8±0.8 | 11.2±1.4 |
| | W/O CLUSTERING | ✓ | ✓ | ✓ | ✗ | ✓ | ✓ | 27.1±1.5 | 25.0±0.7 | 12.5±0.4 |
| | W/O RECON | ✓ | ✓ | ✓ | ✓ | ✗ | ✓ | 33.0±0.5 | 31.1±0.3 | 17.2±0.2 |
| | W/O OT COUPLING | ✓ | ✓ | ✓ | ✓ | ✓ | ✗ | 38.1±0.6 | 36.5±1.0 | 22.5±0.9 |
| | FULL MODEL | ✓ | ✓ | ✓ | ✓ | ✓ | ✓ | **38.3±1.4** | **37.0±0.3** | **23.2±0.8** |

*Table 12.* Complete sensitivity analysis for all hyperparameters on Scene-15. Parameters are ranked by sensitivity score $S$.

| Parameter | Range | $S$ | Best | $\text{ACC}_{\text{best}}$ |
|---|---|---|---|---|
| $\lambda_g$ | [0.01, 1.0] | 0.082 | 0.072 | 0.404±0.009 |
| $\lambda_r$ | [0.1, 3.16] | 0.052 | 1.0 | 0.395±0.020 |
| $\lambda_c$ | [0.1, 10.0] | 0.050 | 2.68 | 0.399±0.012 |
| ODE Steps | [5, 50] | 0.046 | 10 | 0.400±0.013 |
| Latent Dim | [32, 384] | 0.044 | 32 | 0.395±0.016 |
| Dropout | [0.0, 0.5] | 0.043 | 0.0 | 0.389±0.008 |
| Learning Rate | [1e-4, 1e-2] | 0.034 | 2.2e-3 | 0.392±0.011 |
| $\lambda_t$ | [0.01, 1.0] | 0.022 | 0.01 | 0.394±0.010 |
| Flow Hidden Dim | [128, 512] | 0.019 | 192 | 0.397±0.009 |

step count differs slightly by dataset, performance remains stable for a wide range of integration budgets, validating our theoretical analysis that optimal transport paths enable efficient few-step generation.

**Architecture Parameters.** Both latent dimension and flow hidden dimension exhibit low sensitivity ($S < 0.05$), suggesting that OPTION's performance is robust to architectural choices within reasonable ranges. The optimal latent dimension of 32 is notably smaller than the default value of 128, suggesting potential for computational savings without performance degradation.

**Training Parameters.** Learning rate and dropout show the lowest sensitivity among examined parameters, indicating that standard training practices (learning rate $\approx 10^{-3}$, minimal or no dropout) work well across different configurations.

### I.3. Correlation Analysis

We compute Pearson correlation coefficients between parameter values and ACC to identify monotonic relationships:

- **Negative correlations**: Dropout ($r = -0.98$, $p < 0.001$) shows strong negative correlation, confirming that regularization through dropout is unnecessary for this model.

- **Non-monotonic relationships**: $\lambda_g$ ($r = 0.23$, $p = 0.58$), $\lambda_c$ ($r = 0.49$, $p = 0.22$), and ODE steps ($r = -0.20$, $p = 0.67$) show weak correlations with high p-values, indicating optimal performance at intermediate values rather than extremes.

### I.4. Additional Cross-Dataset Sensitivity

To examine whether the sensitivity conclusions transfer beyond Scene-15, we additionally report the four most sensitive parameters on all six benchmark datasets in Table 14. The sensitivity score $S$ follows the same definition as the main text, and the values are computed from the full sensitivity sweep files under `sensitivity_results`.

Figures 6–10 further provide the full nine-parameter sensitivity curves for the remaining datasets. Together with Figure 5, these plots cover the complete six-dataset sensitivity scope.

## J. Additional Convergence Analysis

We provide the complete multi-seed convergence analysis on all six benchmark datasets: BDGP, COIL-20, CUB, Handwritten, NUS-WIDE, and Scene-15. Each experiment follows the same protocol: 5 independent runs with random seeds 42–46, tracking clustering metrics (ACC, NMI, ARI) and loss components over 100 training epochs.

As shown in Figure 11, all six datasets demonstrate consistent convergence patterns: (i) rapid initial improvement within the first 20–30 epochs; (ii) decreasing loss curves with limited oscillation; (iii) bounded standard deviation bands indicating reproducible optimization across random seeds. These results confirm that OPTION's favorable convergence properties generalize across datasets with varying numbers of views, clusters, and sample sizes.

*Table 13.* ODE-step sensitivity on six benchmark datasets (mean $\pm$ std, 5 runs).

| Dataset | ODE Steps | ACC | NMI | ARI | F1 |
|---|---|---|---|---|---|
| BDGP | 5 | 0.347±0.020 | 0.120±0.033 | 0.091±0.025 | 0.321±0.024 |
| | 7 | 0.400±0.046 | 0.166±0.038 | 0.139±0.045 | 0.373±0.045 |
| | 10 | 0.344±0.034 | 0.103±0.047 | 0.087±0.043 | 0.315±0.043 |
| | 15 | 0.351±0.042 | 0.104±0.047 | 0.085±0.043 | 0.320±0.044 |
| | 20 | 0.331±0.013 | 0.083±0.020 | 0.062±0.008 | 0.303±0.018 |
| | 30 | 0.372±0.064 | 0.131±0.074 | 0.111±0.070 | 0.347±0.074 |
| | 50 | 0.335±0.023 | 0.091±0.036 | 0.069±0.023 | 0.305±0.026 |
| COIL-20 | 5 | 0.788±0.019 | 0.889±0.006 | 0.757±0.013 | 0.784±0.021 |
| | 7 | 0.788±0.019 | 0.889±0.006 | 0.757±0.013 | 0.784±0.021 |
| | 10 | 0.788±0.019 | 0.889±0.006 | 0.757±0.013 | 0.784±0.021 |
| | 15 | 0.788±0.019 | 0.889±0.006 | 0.757±0.013 | 0.784±0.021 |
| | 20 | 0.788±0.019 | 0.889±0.006 | 0.757±0.013 | 0.784±0.021 |
| | 30 | 0.788±0.019 | 0.889±0.006 | 0.757±0.013 | 0.784±0.021 |
| | 50 | 0.788±0.019 | 0.889±0.006 | 0.757±0.013 | 0.784±0.021 |
| CUB | 5 | 0.708±0.019 | 0.724±0.011 | 0.590±0.008 | 0.693±0.021 |
| | 7 | 0.711±0.037 | 0.715±0.015 | 0.584±0.015 | 0.697±0.044 |
| | 10 | 0.671±0.021 | 0.706±0.012 | 0.570±0.011 | 0.650±0.028 |
| | 15 | 0.691±0.038 | 0.715±0.013 | 0.581±0.010 | 0.671±0.049 |
| | 20 | 0.673±0.030 | 0.696±0.011 | 0.563±0.009 | 0.654±0.038 |
| | 30 | 0.686±0.045 | 0.714±0.012 | 0.579±0.015 | 0.665±0.050 |
| | 50 | 0.699±0.019 | 0.707±0.014 | 0.572±0.014 | 0.685±0.020 |
| Handwritten | 5 | 0.891±0.054 | 0.863±0.040 | 0.827±0.065 | 0.888±0.057 |
| | 7 | 0.911±0.040 | 0.878±0.023 | 0.849±0.040 | 0.909±0.042 |
| | 10 | 0.905±0.043 | 0.875±0.027 | 0.841±0.046 | 0.901±0.047 |
| | 15 | 0.929±0.029 | 0.890±0.016 | 0.868±0.028 | 0.928±0.031 |
| | 20 | 0.879±0.057 | 0.860±0.028 | 0.817±0.052 | 0.875±0.061 |
| | 30 | 0.915±0.045 | 0.883±0.027 | 0.857±0.048 | 0.913±0.047 |
| | 50 | 0.905±0.036 | 0.874±0.016 | 0.842±0.030 | 0.904±0.037 |
| NUS-WIDE | 5 | 0.158±0.009 | 0.214±0.010 | 0.059±0.006 | 0.129±0.007 |
| | 7 | 0.157±0.011 | 0.214±0.011 | 0.057±0.007 | 0.126±0.008 |
| | 10 | 0.159±0.009 | 0.214±0.009 | 0.057±0.006 | 0.129±0.007 |
| | 15 | 0.156±0.007 | 0.207±0.010 | 0.056±0.005 | 0.127±0.004 |
| | 20 | 0.154±0.007 | 0.212±0.007 | 0.056±0.005 | 0.124±0.009 |
| | 30 | 0.157±0.006 | 0.211±0.010 | 0.059±0.006 | 0.126±0.006 |
| | 50 | 0.158±0.006 | 0.214±0.005 | 0.060±0.005 | 0.129±0.007 |
| Scene-15 | 5 | 0.394±0.011 | 0.424±0.005 | 0.256±0.008 | 0.366±0.009 |
| | 7 | 0.385±0.012 | 0.420±0.009 | 0.251±0.011 | 0.355±0.011 |
| | 10 | 0.400±0.013 | 0.425±0.007 | 0.257±0.010 | 0.372±0.010 |
| | 15 | 0.382±0.020 | 0.417±0.012 | 0.250±0.013 | 0.357±0.020 |
| | 20 | 0.382±0.004 | 0.417±0.007 | 0.247±0.004 | 0.352±0.004 |
| | 30 | 0.393±0.014 | 0.418±0.012 | 0.251±0.013 | 0.362±0.013 |
| | 50 | 0.387±0.016 | 0.421±0.016 | 0.251±0.014 | 0.359±0.016 |

*Table 14.* Additional sensitivity summaries on six benchmark datasets. For each dataset, the four most sensitive hyperparameters are shown.

| Dataset | Parameter | $S$ | Best | $\text{ACC}_{\text{best}}$ |
|---|---|---|---|---|
| BDGP | Dropout | 0.345 | 0.5 | 0.487±0.011 |
| | $\lambda_r$ | 0.190 | 1 | 0.384±0.046 |
| | $\lambda_g$ | 0.180 | 0.518 | 0.403±0.042 |
| | ODE Steps | 0.172 | 7 | 0.400±0.046 |
| COIL-20 | $\lambda_c$ | 0.213 | 0.72 | 0.817±0.016 |
| | Learning Rate | 0.173 | 0.01 | 0.816±0.029 |
| | $\lambda_t$ | 0.151 | 0.464 | 0.819±0.018 |
| | Flow Hidden Dim | 0.071 | 384 | 0.828±0.027 |
| CUB | Dropout | 0.119 | 0 | 0.729±0.023 |
| | ODE Steps | 0.056 | 7 | 0.711±0.037 |
| | $\lambda_g$ | 0.051 | 0.139 | 0.709±0.032 |
| | Learning Rate | 0.048 | 0.002154 | 0.703±0.038 |
| Handwritten | $\lambda_t$ | 0.277 | 0.215 | 0.949±0.005 |
| | $\lambda_c$ | 0.168 | 1.39 | 0.950±0.011 |
| | Learning Rate | 0.112 | 2e-4 | 0.934±0.036 |
| | $\lambda_g$ | 0.093 | 0.01 | 0.947±0.009 |
| NUS-WIDE | Latent Dim | 0.111 | 256 | 0.169±0.006 |
| | Learning Rate | 0.066 | 0.001 | 0.165±0.004 |
| | $\lambda_r$ | 0.060 | 0.1778 | 0.163±0.004 |
| | Dropout | 0.059 | 0.2 | 0.159±0.010 |
| Scene-15 | $\lambda_g$ | 0.082 | 0.072 | 0.404±0.009 |
| | $\lambda_r$ | 0.052 | 1 | 0.395±0.020 |
| | $\lambda_c$ | 0.050 | 2.68 | 0.399±0.012 |
| | ODE Steps | 0.046 | 10 | 0.400±0.013 |

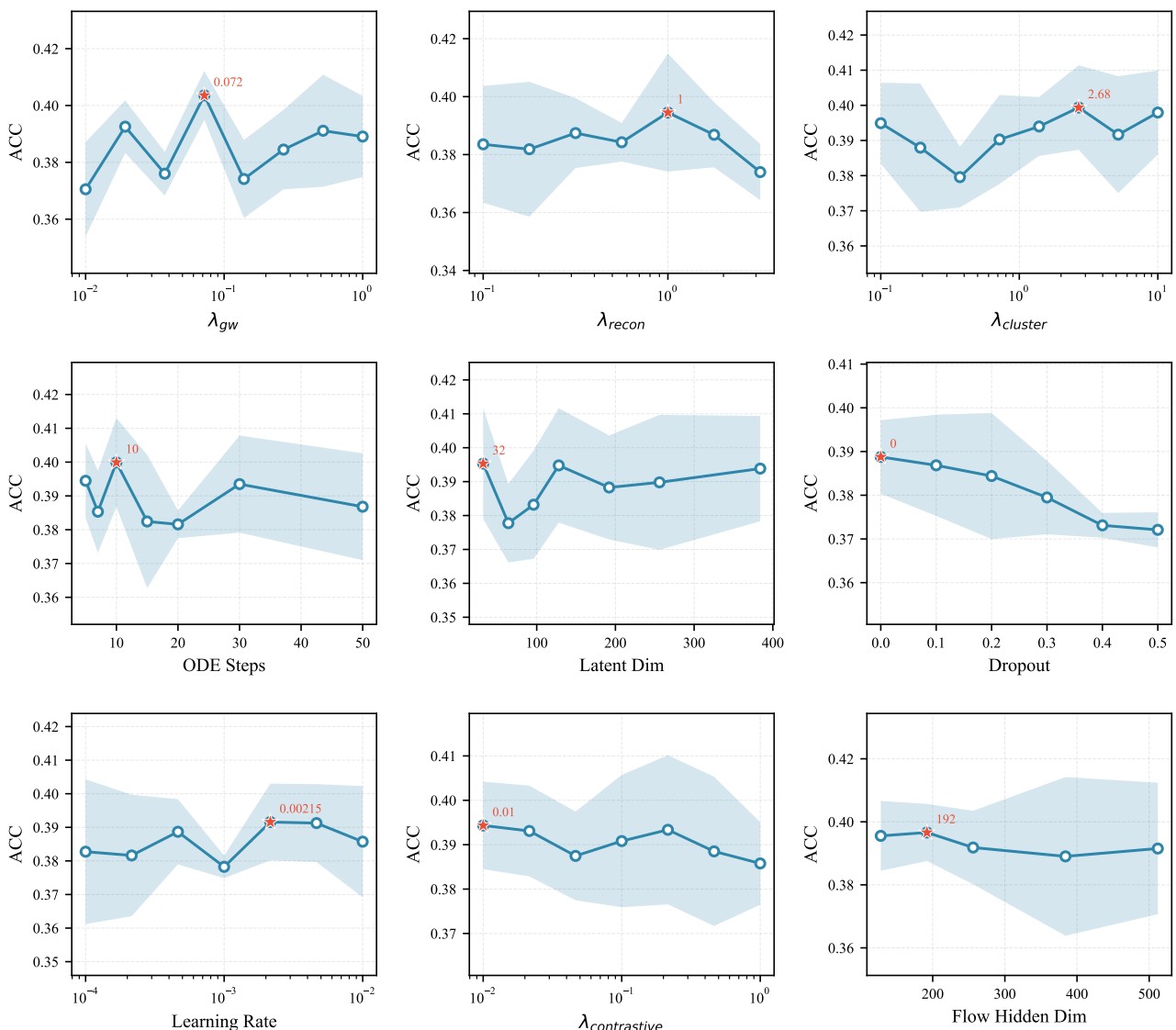

*Figure 5.* Complete sensitivity analysis for all nine hyperparameters on Scene-15. Each subplot shows ACC as a function of parameter value with $\pm 1$ std error bands. Red stars indicate optimal values.

## K. Representative t-SNE Visualizations

This appendix provides t-SNE visualizations of latent representations before and after training for six benchmark datasets, demonstrating the convergence of OPTION's learned representations.

## Limitations

While OPTION demonstrates strong performance, two limitations remain. First, the pairwise structural loss scales as $O(V^2 \cdot B^2)$ with the number of views $V$ and batch size $B$, which may become expensive for many-view scenarios (e.g., $V > 10$). Second, although our sensitivity study shows low sensitivity on the evaluated datasets, the presence of multiple loss weights and architectural choices may still require validation on new datasets.

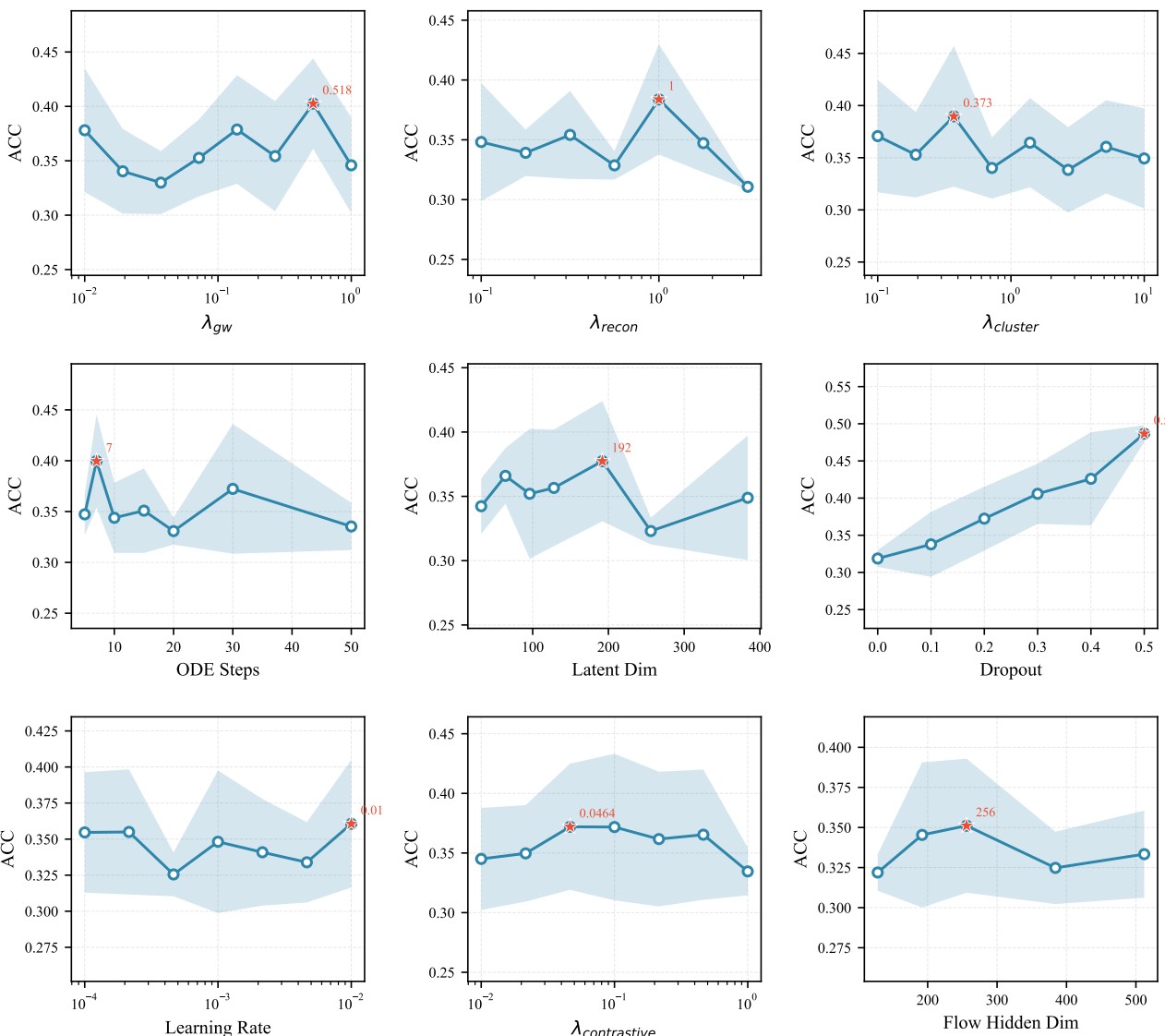

*Figure 6.* Complete sensitivity analysis for all nine hyperparameters on BDGP. Each subplot shows ACC as a function of parameter value with ±1 std error bands.

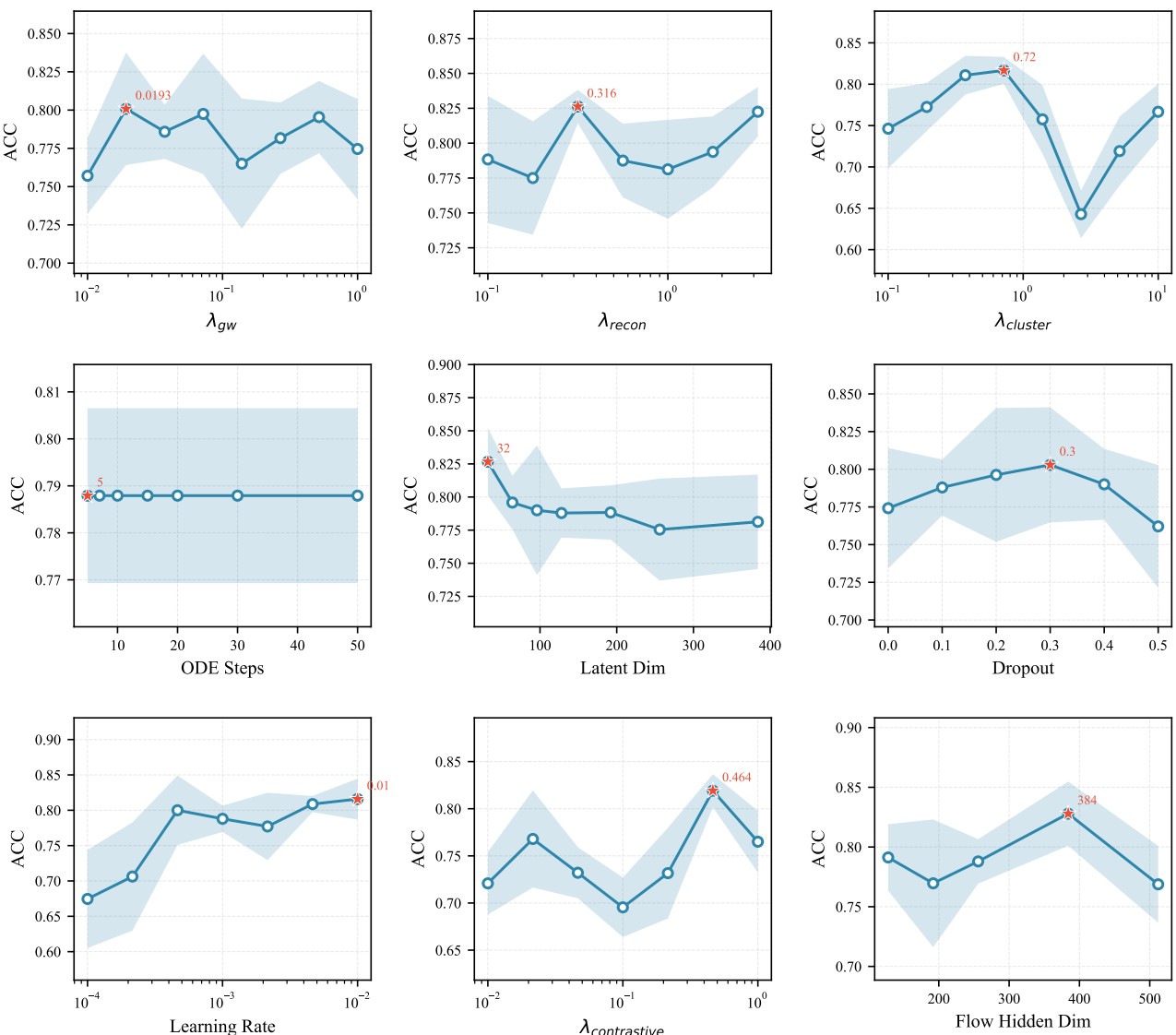

*Figure 7.* Complete sensitivity analysis for all nine hyperparameters on COIL-20. Each subplot shows ACC as a function of parameter value with $\pm 1$ std error bands.

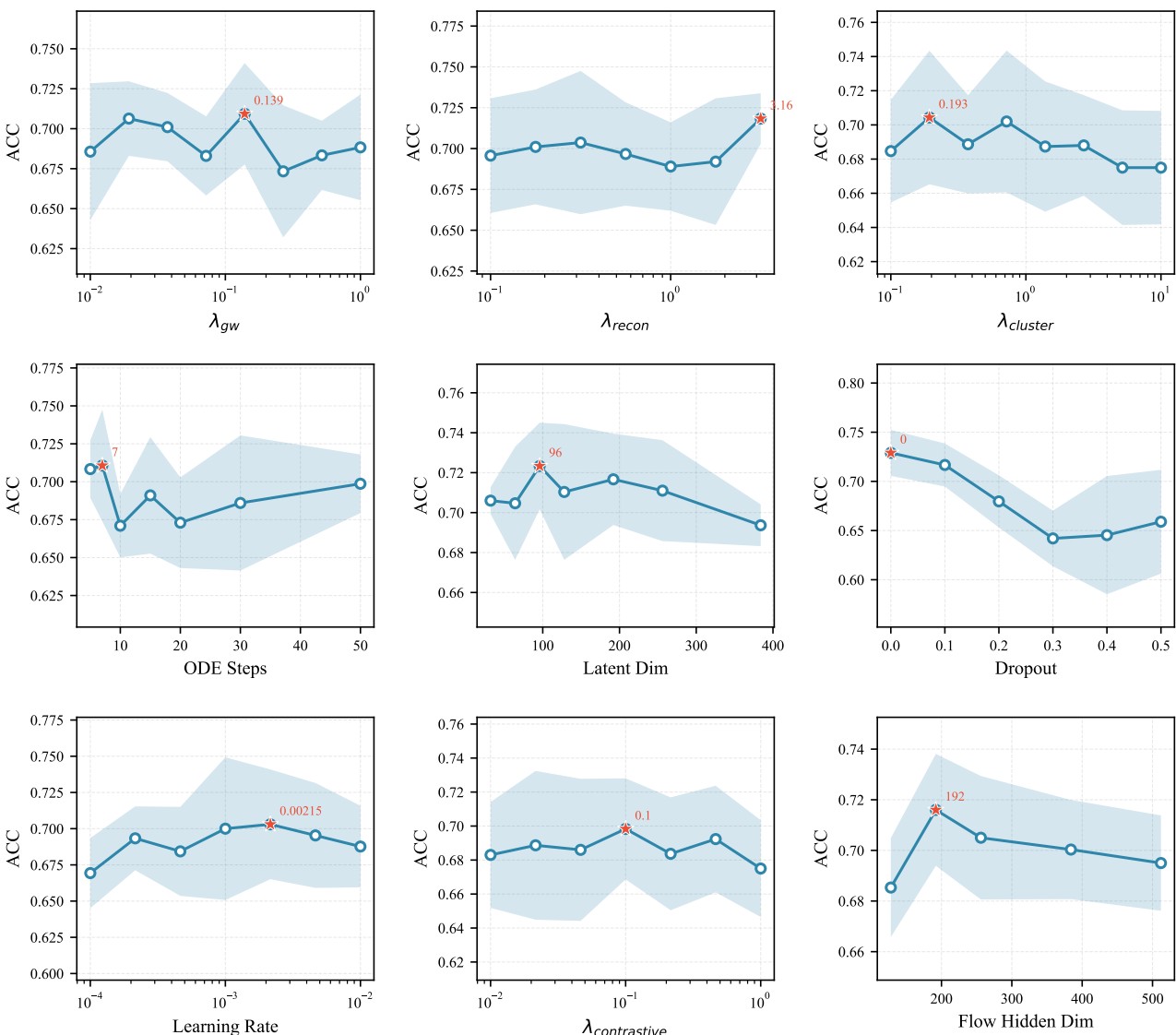

*Figure 8.* Complete sensitivity analysis for all nine hyperparameters on CUB. Each subplot shows ACC as a function of parameter value with ±1 std error bands.

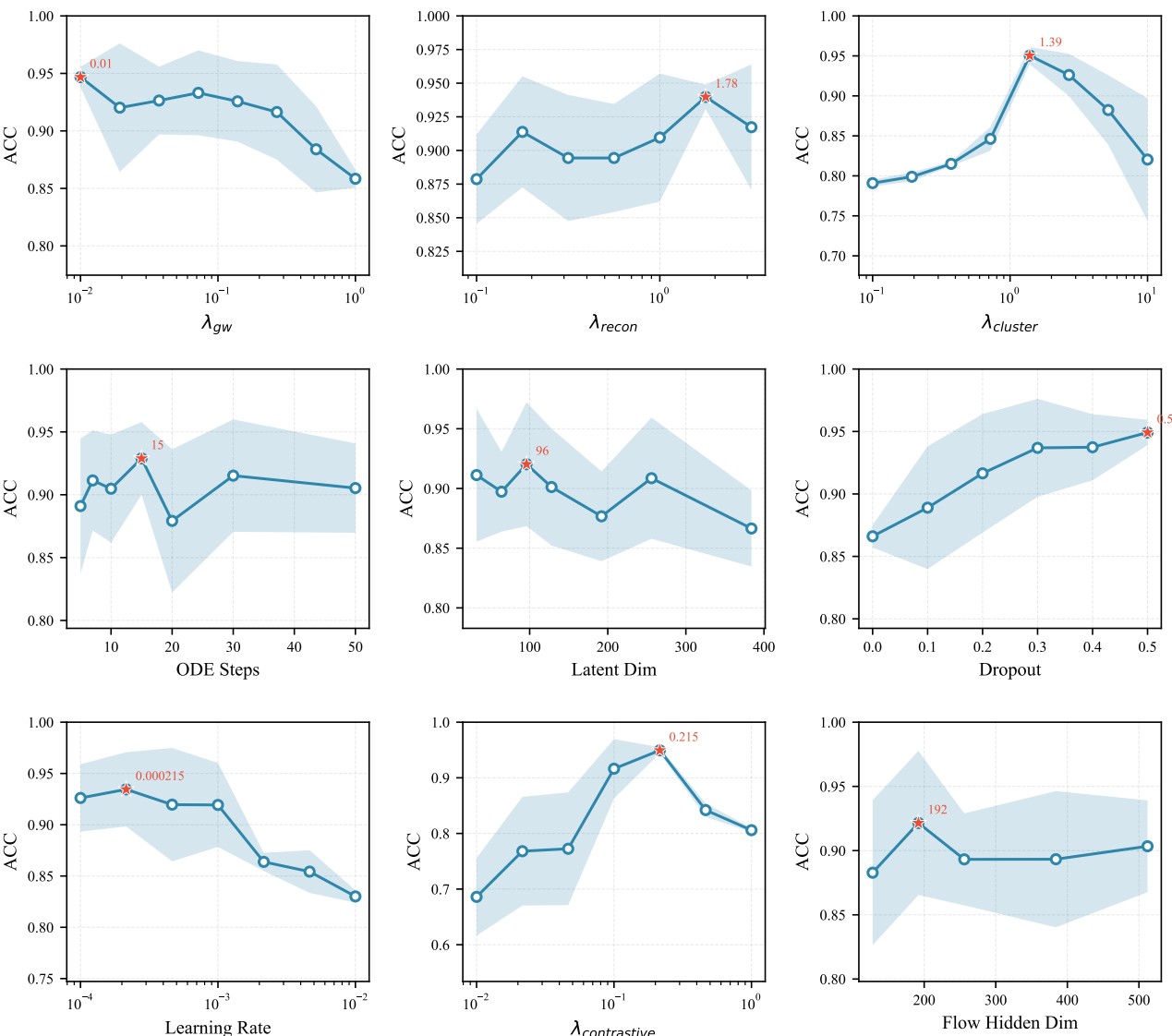

*Figure 9.* Complete sensitivity analysis for all nine hyperparameters on Handwritten. Each subplot shows ACC as a function of parameter value with $\pm 1$ std error bands.

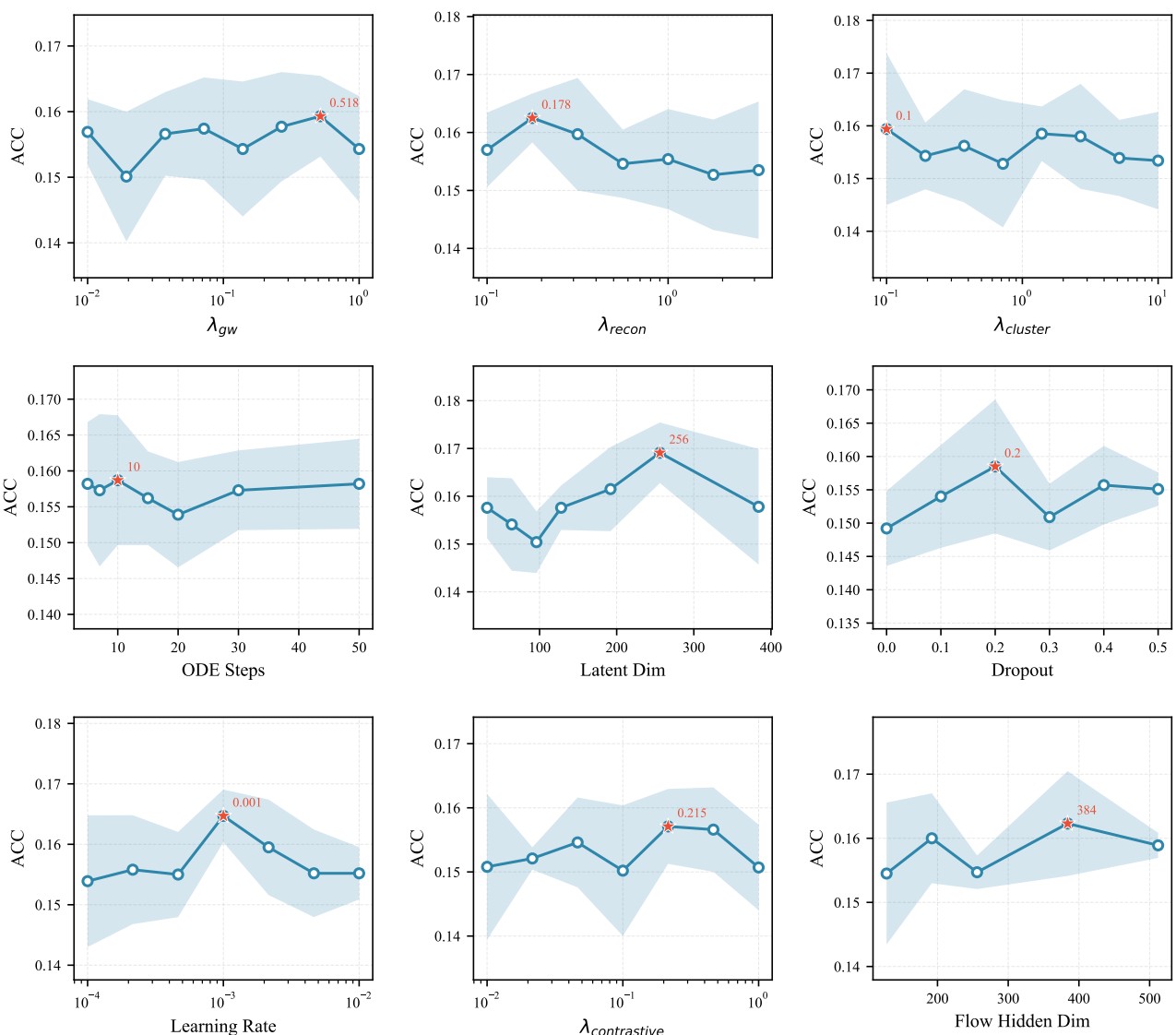

*Figure 10.* Complete sensitivity analysis for all nine hyperparameters on NUS-WIDE. Each subplot shows ACC as a function of parameter value with $\pm 1$ std error bands.

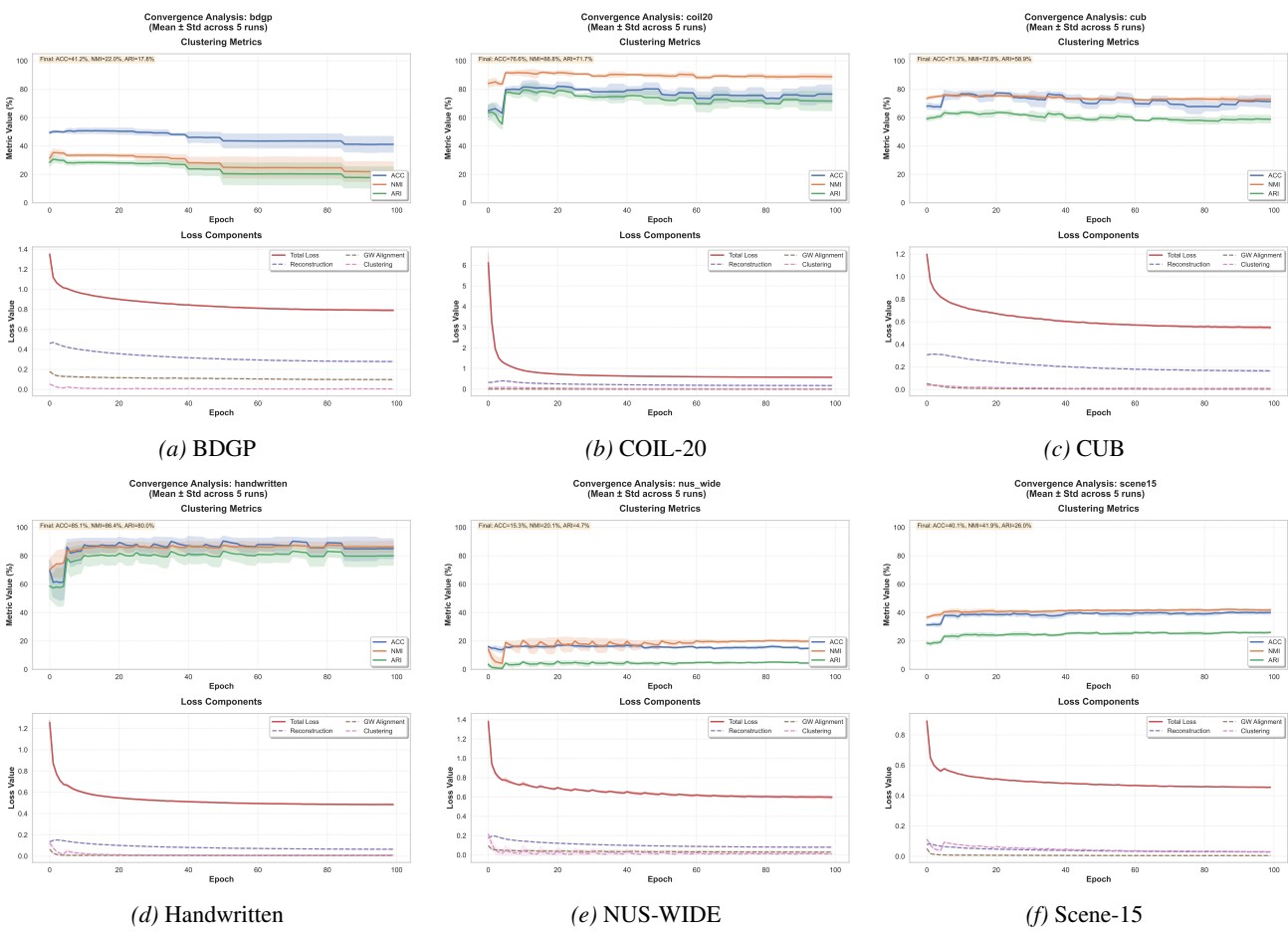

*(a)* BDGP     *(b)* COIL-20     *(c)* CUB

*(d)* Handwritten     *(e)* NUS-WIDE     *(f)* Scene-15

*Figure 11.* Six-dataset multi-seed convergence analysis (mean ± std, 5 runs). Across datasets with different scales, view counts, and cluster structures, OPTION exhibits stable metric trajectories and reproducible loss optimization.

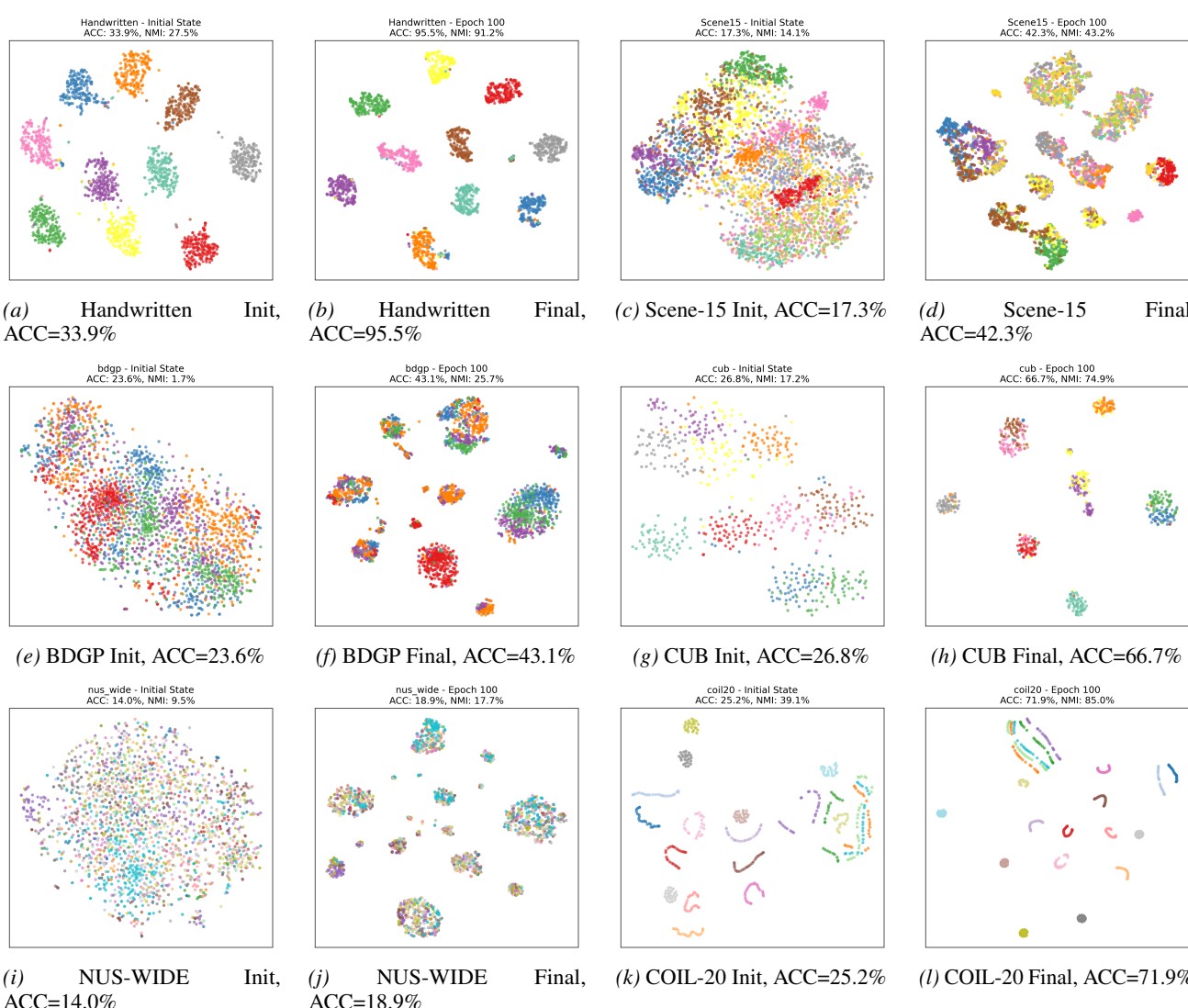

*(a)* Handwritten Init, ACC=33.9%

*(b)* Handwritten Final, ACC=95.5%

*(c)* Scene-15 Init, ACC=17.3%

*(d)* Scene-15 Final, ACC=42.3%

*(e)* BDGP Init, ACC=23.6%

*(f)* BDGP Final, ACC=43.1%

*(g)* CUB Init, ACC=26.8%

*(h)* CUB Final, ACC=66.7%

*(i)* NUS-WIDE Init, ACC=14.0%

*(j)* NUS-WIDE Final, ACC=18.9%

*(k)* COIL-20 Init, ACC=25.2%

*(l)* COIL-20 Final, ACC=71.9%

*Figure 12.* t-SNE visualization of latent representations before and after training across six benchmark datasets. The clusters become significantly more compact and separable after training.

