# OpenReview forum: "OPTION: Optimal Transport–Guided Flow Matching for Incomplete and Unaligned Multi-View Clustering"
_ICML.cc/2026/Conference — ICML 2026 regular_

### Official Review · Reviewer_scLv · 2026-03-03

**Soundness:** 3
**Presentation:** 3
**Significance:** 3
**Originality:** 3
**Overall Recommendation:** 5
**Confidence:** 5

**Summary:**

This work introduces a new approach to multi-view clustering, addressing issues of missing and misaligned data. By leveraging flow matching, the proposed OPTION method enables fast and deterministic inference. The OPTION method also employs a Gromov-Wasserstein loss to enforce geometric consistency across views without needing alignment, and integrates contrastive learning for aligned data. Extensive experiments demonstrate its superior performance compared to current methods.

**Compliance With Llm Reviewing Policy:**

Affirmed.

**Final Justification:**

I think all my concerns have been addressed well, so I decide to raise my socre.

**Key Questions For Authors:**

See Weaknesses.

**Limitations:**

A limitation of the method is the need to tune a large number of parameters, making optimization more complex.

**Strengths And Weaknesses:**

Strengths:
1. The motivation presented in the introduction is clear and easy to follow.
2. The idea of using Flow Matching to generate missing information in multi-view learning is an interesting and novel approach.
3. The optional contrastive regularization for aligned settings adds an extra layer of insight into cross-view consistency.

Weaknesses:
1. Some of the technical terms seem a bit abrupt, such as "10-100× faster inference." The significance of this claim in the context of multi-view clustering should be better explained.
2. There are some formatting inconsistencies, such as references to the appendix—sometimes as "Appendix" and other times as "Appendix D."
3. Some sections, like 4.2.1 "INCOMPLETE MULTI-VIEW CLUSTERING," feel somewhat disjointed. It would be more logical to introduce Table 1 before discussing the data.
4. The choice of Flow Matching over Diffusion needs further justification.
5. The explanation of the optional contrastive learning mechanism is insufficient.
6. Table 3 and Figure 3 use different dataset. This inconsistency undermines the credibility of the experimental evaluation.

---

> ### Author Rebuttal · Authors · 2026-03-30
>
> Thank you for the constructive comments and positive assessment. We address each point below.
>
> > W1: Speed claim is abrupt.
>
> Thanks for this constructive comment. We agree that speed claims should be contextualized for practical MVC usage. Here “10-100× faster inference” refers to the reduction in iterative denoising steps when comparing diffusion-based sampling to our low-step ODE integration in flow matching under the same protocol. As reported in Table 4, the measured speedup has an average of 8.0\\texttimes, with a maximum of 10-100\\texttimes. We will add a clear and complete discussion of its significance for multi-view clustering in the revised manuscript.
>
> > W2: Formatting inconsistency (Appendix vs Appendix D).
>
> Thank you for identifying this editorial issue. The reason is that we initially had multiple appendices but later merged them into one. We will standardize all appendix references and perform a full consistency pass for figure/table/section citations.
>
> > W3: Result sections feel disjointed.
>
> Thank you for this structural suggestion. We will reorder each result section as: setup (e.g., missing-rate protocol), followed by the main table, then key observations. The same structure will be applied to Sec. 4.2.2 (Unaligned MVC) for coherence.
>
> We will also add one transition sentence before each table, so readers can map conditions to outcomes without back-and-forth reading.
>
> > W4: Why flow matching over diffusion.
>
> Thanks for this insightful comment. We choose flow matching for the following reasons:
>
> 1. Diffusion usually needs $T = 100 \sim 1000$ denoising steps; our OT-guided deterministic ODE uses $N = 10$, giving about $8\times$ average speedup (Table 4).
> 2. Diffusion is stochastic during sampling, while flow matching is deterministic under fixed noise/conditioning, which is preferable for reproducible clustering.
> 3. OT-guided near-linear paths produce smoother vector fields, faster convergence, and fewer integration errors at coarse step sizes.
> 4. Flow matching directly regresses target velocity via MSE, avoiding complex noise schedules; our convergence curves (Fig. 4) are smooth and stable across seeds.
>
> We will add a dedicated paragraph in the Method section to more clearly articulate these advantages in the revised manuscript.
>
> > W5: Contrastive mechanism is underexplained.
>
> Thanks for this constructive comment. Optional contrastive loss $\mathcal{L}_{con}$ (InfoNCE) is enabled only in aligned IMVC and disabled in UMVC ($\lambda_t = 0$).
>
> For aligned data, positive pairs are same-sample cross-view latents $(\mathbf{z}_i^{(u)}, \mathbf{z}_i^{(v)})$; other samples are negatives:
>
> $\mathcal{L}\_{con} = -\frac{1}{N} \sum\_i \log \frac{\exp(\text{sim}(\mathbf{z}\_i^{(u)}, \mathbf{z}\_i^{(v)}) / \tau)}{\sum\_{j=1}^{N} \exp(\text{sim}(\mathbf{z}\_i^{(u)}, \mathbf{z}\_j^{(v)}) / \tau)}$
> where $\text{sim}(\cdot, \cdot)$ is cosine similarity and $\tau$ is temperature.
>
> While GW provides structure-level alignment, InfoNCE adds instance-level alignment. Their combination improves aligned MVC. In UMVC, true positive pairs are unavailable, so GW alone is used. This modular switch is a core design of OPTION. Table 3 shows removing $\mathcal{L}_{con}$ causes the largest ACC drop (12.2\%), confirming its importance in aligned settings. We will expand the description of the contrastive mechanism in the revised manuscript.
>
> > W6: Dataset inconsistency between Table 3 and Figure 3.
>
> Thank you for this insightful observation. We clarify the reasoning behind this choice: Table 3 (Ablation study) uses Scene-15 and Figure 3 (Sensitivity analysis) uses Handwritten to demonstrate that the observed trends are consistent across different datasets. However, we acknowledge that using the same dataset for both analyses would improve coherence. In the revised manuscript, we will ensure that both the ablation study and sensitivity analysis use the same dataset(s) for better consistency. We will also add additional ablation and sensitivity results on other datasets in the Appendix for completeness.
>
> > Limitation comment: Too many parameters to tune.
>
> We acknowledge that the requirement to configure multiple loss weights and hyperparameters is a valid limitation of our framework, as it expands the potential search space. However, the sensitivity analysis (provided in the main text and appendix) indicates that the model exhibits a relatively acceptable level of sensitivity to these choices.  To alleviate the tuning burden, we have identified a set of robust practical defaults that perform consistently well across all evaluated scenarios:
> $\lambda_g \in [0.05, 0.2]$,
> $\lambda_c \in [1.0, 3.0]$,
> $\lambda_r \in [0.5, 1.5]$,
> $\lambda_t = 0.3$ (aligned) or $0$ (unaligned),
> $N = 10$ ODE steps.
>
> Besides, OPTION's fast training convergence makes grid search or Bayesian optimization feasible within reasonable time budgets. We will add a practical deployment guide with recommended default hyperparameters in the revised manuscript.

---

> > ### Author Rebuttal · Reviewer_scLv · 2026-04-02
> >
> > The authors answered all my questions.

---

### Official Review · Reviewer_dDKM · 2026-03-10

**Soundness:** 3
**Presentation:** 3
**Significance:** 3
**Originality:** 3
**Overall Recommendation:** 4
**Confidence:** 5

**Summary:**

This paper introduces OPTION (OPtimal Transport–GuIded flOw MatchiNg), a method designed to tackle challenges in multi-view clustering due to missing views and cross-view misalignment. OPTION uses conditional flow matching to learn transport paths for imputing missing data, preserving the underlying manifold structure. To align geometric structures across views without exact alignment, it employs a Gromov-Wasserstein loss. Additionally, an optional contrastive regularization is applied to improve cross-view consistency when views are aligned.

**Compliance With Llm Reviewing Policy:**

Affirmed.

**Final Justification:**

My concerns have now been resolved, and I maintain my positive rating.

**Key Questions For Authors:**

1. The conditional extension of Flow Matching in unaligned settings is not well-explained. More details on how the cluster centroid is determined and its impact on generation are required.
2. The proposed method is faster than diffusion-based approaches, but how does its computational speed compare with other methods?

**Limitations:**

N.A.

**Strengths And Weaknesses:**

Strengths：

The proposed OPTION method is technically sound, demonstrating advantages over prior work. Extensive experiments show that it outperforms state-of-the-art methods across ideal, incomplete, and misaligned scenarios, highlighting both its robustness and effectiveness in handling challenging multi-view data.

Weaknesses：

1. The authors mention that diffusion-based IMVC relies on iterative refinement to reconstruct missing views, which limits efficiency in dynamic or large-scale multi-view scenarios. However, there is no explanation of how the proposed method overcomes this limitation.
2. The "Related Work" section does not provide a comprehensive overview of the current research. Some important works have been omitted.
3. The paper does not explain how the kernel matrix parameter lambda is adjusted.
4. In the Gromov-Wasserstein Structural Alignment module, structural alignment is used, but errors in alignment may lead to incorrect information. The paper does not explain how such issues are handled or mitigated. How does the method ensure robustness against erroneous structures?
5. Several key components in Eq. (5) are not defined, which affects clarity and readability.

---

> ### Author Rebuttal · Authors · 2026-03-30
>
> Thank you for the careful and constructive review. We respond point by point.
>
> > W1: Efficiency not sufficiently addressed.
>
> We thank the reviewer for this insightful comment. The efficiency gap originates from a fundamental shift in the generative paradigm. Diffusion-based IMVC reconstructs one missing view through stochastic denoising with typically $T=100 \sim 1000$ network evaluations. OPTION uses conditional flow matching with straight OT-guided paths in a deterministic CNF, so it needs far fewer integration steps. This is exactly why OPTION is on average $8.0\times$ faster than DCG across five datasets (Table 4).
>
> > W2: Related work is not comprehensive.
>
> We will comprehensively update the Related Work section by adding recent surveys and key representative IMVC/UMVC studies.
>
> > W3: Kernel parameter adjustment is unclear.
>
> Thanks for this constructive feedback. We set $\gamma$ adaptively by median heuristic per mini-batch:
>
> $$\gamma = \frac{1}{2\sigma_{\mathrm{med}}^{2}}, \quad \text{where } \sigma_{\mathrm{med}} = \mathrm{median}\left( \{ \|\mathbf{z}\_{i} - \mathbf{z}\_{j}\| \}_{i < j} \right)$$
>
> So $\gamma$ follows latent scale automatically, without manual tuning. In early training (coarser latent space), pairwise distances are larger and $\gamma$ is smaller; later, latent features become compact and $\gamma$ increases. This keeps kernel variation informative and GW calibration stable across all three phases.
>
> > W4: Robustness to structural alignment errors is unclear.
>
> We are grateful to the reviewer for highlighting this point. Our framework ensures robust fault tolerance against alignment errors through three mechanisms:
>
> 1. GW is computed on mini-batches ($B$), not globally, reducing influence of large-scale misalignment noise.
> 2. We use soft kernel similarities rather than hard matching(e.g., Hungarian algorithm), so small latent perturbations only cause small kernel changes, preventing catastrophic error propagation.
> 3. GW is jointly optimized with $\mathcal{L}\_{rec}$, $\mathcal{L}\_{clu}$, and flow loss, where reconstruction and clustering regularize geometry and prevent GW from drifting to spurious structures.
>
> By intentionally avoiding hard one-shot correspondences in favor of soft structural regularization, our method remains highly robust. We will detail this design in the revision.
>
> > W5: Undefined terms in Eq. (5).
>
> Thank you for spotting this. We apologize for the oversight. In Eq. (5),
> $\mathcal{L}\_{tot} = \mathcal{L}\_{CFM} + \lambda\_g \mathcal{L}\_{GW} + \lambda\_c \mathcal{L}\_{clu} + \lambda\_r \mathcal{L}\_{rec} + \lambda\_t \mathcal{L}\_{con}$.
> The first four terms are defined earlier (Eq. 2, Eq. 3, Sec. 3.1, Sec. 3.5), but we agree that $\mathcal{L}\_{con}$ and the hyperparameters $\lambda\_g, \lambda\_c, \lambda\_r, \lambda\_t$ should be defined in place. We will add concise inline definitions at Eq. (5). We will perform a notation consistency check so symbols in equations, method text, and algorithm descriptions are aligned.
>
> > KQ1: Conditional design in unaligned setting and centroid role.
>
> Thanks for this insightful question. In aligned IMVC, conditioning uses sample-level consensus:
> $\mathbf{c}\_i = \frac{1}{|\mathcal{V}\_{avail}^{(i)}|} \sum\_{v \in \mathcal{V}\_{avail}^{(i)}} \mathbf{z}\_i^{(v)}$.
>
> In UMVC, cross-view one-to-one correspondence is unknown, so cross-view averaging is invalid. We instead use cluster-prototype conditioning. For each view representation $\mathbf{z}\_i^{(v)}$, soft assignment is
> $q\_{ik}^{(v)} = \text{softmax}(\tilde{\mathbf{z}}\_i^{(v)\top} \tilde{\boldsymbol{\mu}}\_k / \tau)$,
> and condition is
> $\mathbf{c}\_{i}^{(v)} = \boldsymbol{\mu}\_{k^{\ast}} \text{ with } k^{\ast} = \arg\max\_{k} q\_{ik}^{(v)}$.
> So even without correspondences, generation remains semantically tied to cluster identity.
>
> > KQ2: Speed beyond diffusion baselines.
>
> Table 4 highlights DCG to isolate paradigm difference and compare flow-matching directly with diffusion-based method. For broader context, we also report runtime against non-diffusion SOTA. The time comparison here is based on wall-clock timing under the same benchmarking protocol, as opposed to the step-based comparison between OPTION and DCG in Table 4.
>
> | Method | bdgp(ms) | coil20(ms) | cub(ms) | handwritten(ms) | scene15(ms) |
> |---|---:|---:|---:|---:|---:|
> | OPTION | 2862.9184 | 6760.2645 | 2449.9312 | 3514.8084 | 6585.6697 |
> | MRG-UMC (TNNLS25) | 6765.3933 | 13294.9724 | 3964.0013 | 18035.3354 | 6111.8273 |
> | CANDY (NeurIPS24) | 12899.6416 | 16644.1081 | 5687.8647 | 44773.0678 | 11957.7698 |
> | SURE (TPAMI22) | 15843.1956 | 37692.4747 | 15358.5992 | 14551.9683 | 15038.1452 |
>
> These numbers show that OPTION is consistently competitive in runtime and often substantially faster across datasets. In summary, the table supports our efficiency claim beyond diffusion baselines under a unified setting. We will include the comprehensive speed comparison in the revised manuscript.

---

> > ### Author Rebuttal · Reviewer_dDKM · 2026-04-02
> >
> > Thanks for the author's response. My concerns have now been resolved.

---

### Official Review · Reviewer_KrbS · 2026-03-10

**Soundness:** 3
**Presentation:** 3
**Significance:** 3
**Originality:** 4
**Overall Recommendation:** 4
**Confidence:** 5

**Summary:**

This paper leverages a flow matching mechanism to impute missing views and applies a Gromov-Wasserstein loss to align geometric structures across views without requiring exact correspondences. Its modular design activates contrastive learning for aligned data while relying on GW-based structural alignment for unaligned data. Experiments on four benchmark datasets demonstrate superior clustering performance and faster inference.

**Compliance With Llm Reviewing Policy:**

Affirmed.

**Key Questions For Authors:**

Please refer to the weaknesses section.

**Limitations:**

The mechanism for obtaining learnable centroids lacks clarity.

**Strengths And Weaknesses:**

Strengths:
1. By combining flow matching for missing views with Gromov-Wasserstein alignment for unaligned data, the method robustly improves clustering across multiple benchmarks.
2. Comprehensive experiments and visual analyses effectively support the method’s claims.
Weaknesses:
1. The physical meaning of the kernel matrix K in Equation 1 is not clearly explained, and the dimension B is undefined, creating ambiguity regarding their roles.
2. Section 3.4, "Conditional Flow Matching for Imputation," lacks clarity, particularly regarding the application of optimal transport, making it difficult to understand.
3. Some formulas are presented inconsistently, such as the mix of uppercase and lowercase notations for the loss terms in Equation 5. Standardizing the notation would improve clarity.
4. The setup for the dataset's incompleteness and misalignment is vague, and the manuscript does not provide sufficient details on how these issues were addressed.
5. The parameter sensitivity analysis is conducted on only one dataset, lacking evaluation on other datasets.
6. Why is the data in both Table 1 and Table 2 not entirely consistent, despite both presenting ideal multi-view data, and what is the cause of this discrepancy? Is it due to different preprocessing or other factors?

---

> ### Author Rebuttal · Authors · 2026-03-31
>
> We sincerely thank you for your thoughtful review and constructive suggestions. We address each point below.
>
> > W1: Physical meaning of $\mathbf{K}$ and undefined $B$.
>
> Thank you for identifying this clarity gap. In Eq. 1, $\mathbf{K}^{(v)} \in \mathbb{R}^{B \times B}$ is an RBF kernel over view-$v$ latents, and $B$ is mini-batch size.
>
> Each element
> $K\_{ij}^{(v)} = \exp(-\gamma \|\mathbf{z}\_i^{(v)} - \mathbf{z}\_j^{(v)}\|^2)$
> measures pairwise similarity: values near 1 imply close latent neighbors; near 0 imply far samples. We set
> $\gamma = 1 / \text{median}(\{\|\mathbf{z}\_i - \mathbf{z}\_j\|^2 : i \neq j\})$.
> So $\mathbf{K}^{(v)}$ encodes local manifold geometry in latent space. We will define $B$ explicitly and strengthen this physical interpretation in the revision. To avoid ambiguity, we will place this definition at first appearance in Eq. 1 and keep the same wording in the method text.
>
> > W2: Sec. 3.4 clarity on CFM and OT.
>
> Thanks for this constructive comment. We use OT-guided conditional paths (Lipman et al., 2023), not arbitrary interpolation. Path and target field are:
> $\mathbf{z}\_t = (1-t)\mathbf{z}\_0 + t\mathbf{z}\_1$,
> $u\_t = \mathbf{z}\_1 - \mathbf{z}\_0$,
> with $\mathbf{z}\_0 \sim \mathcal{N}(0, I)$.
> The training loss is
> $\mathcal{L}\_{CFM} = \mathbb{E}\_{t, \mathbf{z}\_0, \mathbf{z}\_1, \mathbf{c}} [\|v\_\theta(\mathbf{z}\_t, t, \mathbf{c}) - (\mathbf{z}\_1 - \mathbf{z}\_0)\|^2]$.
>
> Benefits are fewer ODE steps (10 in our setting) for efficient inference, and smoother vector fields with better training stability than curved/noisy paths. We will rewrite Sec. 3.4 for clearer OT motivation and add a compact OT-to-CFM bridge sentence to connect the path definition and the practical low-step ODE setting.
>
> > W3: Inconsistent notation in Eq. 5.
>
> Thank you for this insightful observation. We will unify notation as:
> $\mathcal{L}\_{\text{flow}}$, $\mathcal{L}\_{\text{gw}}$, $\mathcal{L}\_{\text{clu}}$, $\mathcal{L}\_{\text{rec}}$, $\mathcal{L}\_{\text{con}}$,
> and define $\lambda\_g, \lambda\_c, \lambda\_r, \lambda\_t$ with defaults immediately after Eq. 5. The same naming will be used consistently in ablation tables and algorithm descriptions in the revised manuscript.
>
> > W4: Setup of incompleteness and misalignment is vague.
>
> Thanks for this helpful comment. We will clarify protocol details.
>
> Incomplete setting: each view of each sample is independently removed with probability $r_{m} \in \\{0\text{\\%}, 10\text{\\%}, 30\text{\\%}, 50\text{\\%}, 70\text{\\%}\\}$, while ensuring every sample keeps at least one view.
>
> Unaligned setting: for $r_{u} \in \\{0\text{\\%}, 20\text{\\%}, 40\text{\\%}, 60\text{\\%}\\}$, we randomly choose $r_{u} \times N$ samples in one view and shuffle their indices; remaining $(1-r_{u}) \times N$ samples keep correct correspondence. We will mirror this protocol description in both main text and appendix to avoid cross-section inconsistencies.
>
> > W5: Sensitivity analysis only on one dataset.
>
> Thanks for this insightful suggestion. Scene-15 was chosen for challenge and heterogeneity. We agree that evaluating on additional datasets would strengthen our claims. Here we only report two additional datasets because of limited space, where only four parameters with highest sensitivity are reported.
>
> CUB:
> | Parameter | Range | S | Best | ACC_best |
> |---|---|---:|---:|---|
> | Dropout | [0, 0.5] | 0.119 | 0 | 0.729±0.023 |
> | ODE Steps | [5, 50] | 0.056 | 7 | 0.711±0.037 |
> | lambda_gw | [0.01, 1] | 0.051 | 0.139 | 0.709±0.032 |
> | Learning Rate | [0.0001, 0.01] | 0.048 | 0.002154 | 0.703±0.038 |
>
> NUSWIDE:
> | Parameter | Range | S | Best | ACC_best |
> |---|---|---:|---:|---|
> | Latent Dim | [32, 384] | 0.111 | 256 | 0.169±0.006 |
> | Learning Rate | [0.0001, 0.01] | 0.066 | 0.001 | 0.165±0.004 |
> | lambda_recon | [0.1, 3.162] | 0.060 | 0.1778 | 0.163±0.004 |
> | Dropout | [0, 0.5] | 0.059 | 0.2 | 0.159±0.010 |
>
> where S is the sensitivity score defined in the main text. Overall, while a few parameters exhibit moderate sensitivity (S > 0.1), the variations remain manageable and won't trigger severe instability.
>
> > W6: Table 1 vs Table 2 mismatch at 0%.
>
> Thanks for this insightful observation. The slight differences in 0% columns arise from different random seeds and setting-specific tuning pipelines. We will clarify this in the manuscript.
>
> > Limitation comment: Learnable centroids are unclear.
>
> Thank you for this constructive comment. Centroids $\\{\boldsymbol{\mu}\_k\\}\_{k=1}^K$ are obtained in two stages.
>
> 1. After Phase-1 encoder-decoder pretraining, we run K-Means on concatenated latent representations $\\{\mathbf{z}\_i^{(v)}\\}$ to initialize centroids.
> 2. In end-to-end training, these centroids are learnable parameters updated by gradient descent through
> $\mathcal{L}\_{clu} = \text{KL}(P \| Q)$ (Sec. 3.5).
>
> We will make this procedure explicit in the revised version. We will also state where centroid initialization happens in the training pipeline and when centroid updates are activated.

---

> > ### Author Rebuttal · Reviewer_KrbS · 2026-04-02
> >
> > After reviewing all the comments and rebuttals, I will keep my score.

---

### Official Review · Reviewer_FEgi · 2026-03-12

**Soundness:** 4
**Presentation:** 3
**Significance:** 4
**Originality:** 4
**Overall Recommendation:** 4
**Confidence:** 5

**Summary:**

The paper proposes a novel and unified framework (OPTION) for multi-view clustering that addresses both incomplete and unaligned data. OPTION employs conditional flow matching to learn deterministic
transport paths for missing-view imputation, enabling stable manifold-preserving recovery and
more discriminative representations. The key contributions include flow matching for efficient deterministic inference with lower latency compared to diffusion-based methods, a Gromov-Wasserstein structural coupling loss to enforce cross-view geometric consistency without requiring sample correspondences, and a modular architecture that adapts contrastive learning for aligned data and GW-based alignment for unaligned scenarios.

**Compliance With Llm Reviewing Policy:**

Affirmed.

**Final Justification:**

The authors addressed my main concerns, so I maintain my original positive score.

**Key Questions For Authors:**

Why can flow matching be used to generate missing multi-view data? How does it differ from and outperform other incomplete multi-view clustering methods?

**Limitations:**

The authors have not discussed the limitations or potential societal impact. Addressing issues like scalability, biases, and ethical concerns in sensitive areas would be beneficial.

**Strengths And Weaknesses:**

Strengths:
1.. The method combines conditional flow matching for missing-view imputation with a Gromov-Wasserstein loss for alignment-free fusion. Both components are theoretically justified and implemented correctly.
2. The paper addresses an important problem in multi-view learning: handling incomplete and unaligned views, which is highly relevant to practical applications.
3. Ablation studies validate the contribution of each module, and the experimental design is generally rigorous.
Weaknesses:
1. The explanation of how "conditional semantic information" guides flow matching is insufficient, especially regarding its impact on imputation in aligned vs. unaligned settings and ensuring semantic coherence while allowing independent view processing.
2. Discrepancy in the number of datasets: Tables 1 and 2 mention four datasets, while six are listed in the appendix. Table 4 includes five datasets, and the computational time for the NUSWIDE dataset is missing
3. Tables 1 and 2 omit standard deviation, unlike Table 3. Standardizing data presentation across tables is needed.
4. Appendix figure numbering (Figure 7) is repeated, which should be corrected to avoid confusion.
5. The writing could be improved for better clarity and coherence.
6. The paper claims that 10 steps yield optimal results, but no comparison with other step numbers (e.g., 5 or 15) is provided. Experiments with varying step counts are needed to verify if 10 is truly optimal.

---

> ### Author Rebuttal · Authors · 2026-03-30
>
> Thank you for the positive assessment and constructive suggestions. We address each point concisely.
>
> > W1: Conditional semantics are unclear.
>
> Thanks for this constructive comment. Our conditioning is setting-dependent. In aligned data, we condition on available corresponding views. In unaligned data, one-to-one correspondence is unknown, so cross-view averaging is invalid. We condition on cluster-level semantic prototypes and rely on structural alignment for cross-view consistency. We will add a clearer paragraph in Sec. 3.4.
>
> > W2: Dataset inconsistency and missing NUS-WIDE runtime.
>
> Thank you for identifying this. Appendix lists the full benchmark scope, while Tables 1-2 show four representative datasets (BDGP, Coil20, CUB, NUS-WIDE) due to space. We will make table scope explicit and synchronize main text, tables, and appendix. We will also add the missing NUS-WIDE runtime:
>
> | Dataset | OPTION (ms) | DCG (ms) | Speedup |
> |---------|-------------|----------|---------|
> | NUS-WIDE | 3.00 | 25.29 | 8.4x |
>
> > W3: Missing standard deviations in Tables 1-2.
>
> Thanks for this helpful suggestion. We ran multi-seed experiments and will report mean ± std explicitly.
>
> Experimental setting: each entry repeats full training with different random seeds under the same protocol as the main paper. We report mean ± std over 5 runs (5 seeds) for each missing/unaligned rate and dataset. The original tables reported mean only; the revision will add std and state the run count. Here we only show the updated OPTION results due to space.
>
> Table 1 (Incomplete MVC), OPTION (ACC±std):
>
> | Dataset | 0% | 10% | 30% | 50% |
> |---|---|---|---|---|
> | BDGP | 50.5±0.8 | 50.4±1.0 | 46.6±0.4 | 43.5±1.4 |
> | Coil20 | 88.7±0.7 | 88.4±0.4 | 88.4±0.4 | 87.6±0.8 |
> | CUB | 78.6±3.1 | 79.6±2.1 | 79.3±3.2 | 73.4±2.0 |
> | NUS-WIDE | 19.0±0.7 | 19.4±0.8 | 19.9±0.6 | 19.1±0.5 |
>
> Table 2 (Unaligned MVC), OPTION (ACC±std):
>
> | Dataset | 0% | 20% | 40% | 60% |
> |---|---|---|---|---|
> | BDGP | 50.3±0.3 | 48.3±1.3 | 48.0±3.2 | 46.3±1.0 |
> | Coil20 | 88.7±0.7 | 73.5±1.6 | 66.4±3.4 | 51.5±1.4 |
> | CUB | 79.2±3.3 | 69.5±1.6 | 60.5±1.9 | 53.0±3.0 |
> | NUS-WIDE | 19.6±0.6 | 17.8±0.3 | 15.9±0.7 | 14.1±0.3 |
>
> > W4: Repeated appendix figure numbering.
>
> Thanks for this observation. The two panels actually belong to one figure but were split by layout. We will run a full consistency check of figure/table indices and all in-text references, then fix repeated numbering.
>
> > W5: Writing clarity and coherence.
>
> Thank you for this constructive suggestion. We will improve transitions, define symbols before first use, and tighten technical wording in the revised manuscript.
>
> > W6: Explicit comparison across ODE step numbers.
>
> Thanks for this insightful comment. The analysis exists in the sensitivity study (Fig. 2, ODE Steps), but we agree it should be emphasized more clearly. We will add explicit comparison for multiple steps and explain why 10 is our default quality-efficiency trade-off.
>
> | ODE steps | ACC | NMI | ARI | F1 |
> |---:|---:|---:|---:|---:|
> | 5  | 0.3945 ± 0.0110 | 0.4240 ± 0.0055 | 0.2559 ± 0.0076 | 0.3663 ± 0.0087 |
> | 7  | 0.3853 ± 0.0122 | 0.4202 ± 0.0087 | 0.2507 ± 0.0108 | 0.3546 ± 0.0109 |
> | 10 | 0.3999 ± 0.0131 | 0.4250 ± 0.0072 | 0.2570 ± 0.0103 | 0.3719 ± 0.0102 |
> | 15 | 0.3824 ± 0.0197 | 0.4169 ± 0.0117 | 0.2500 ± 0.0133 | 0.3571 ± 0.0198 |
> | 20 | 0.3816 ± 0.0041 | 0.4172 ± 0.0069 | 0.2474 ± 0.0044 | 0.3520 ± 0.0037 |
> | 30 | 0.3935 ± 0.0144 | 0.4179 ± 0.0117 | 0.2513 ± 0.0132 | 0.3616 ± 0.0125 |
> | 50 | 0.3868 ± 0.0158 | 0.4214 ± 0.0159 | 0.2507 ± 0.0136 | 0.3590 ± 0.0156 |
>
> Performance peaks at 10 steps and does not improve with larger step counts.
>
> Thus, choosing 10 is an evidence-based trade-off.
>
> > KQ: Why flow matching for missing-view generation.
>
> Our goal is reliable semantics plus high efficiency. Flow matching learns deterministic transport from noise to data via Continuous Normalizing Flow(CNF). Given $\mathbf{z}\_0 \sim \mathcal{N}(0, I)$ and condition $\mathbf{c}$ (available-view consensus or cluster centroid), the vector field $v\_\theta$ moves samples along OT-guided paths toward the missing-view latent distribution. This preserves manifold structure, improves consistency, and avoids long stochastic denoising chains.
>
> > Limitation comment: Need limitations and societal impact.
>
> Thank you for this important reminder. We will add a dedicated limitations/broader-impact subsection, including:
> 1. Pairwise GW cost scales as $O(V^2 \cdot B^2)$ and can be expensive for many views ( > 10).
> 2. The presence of multiple loss weights and architectural choices introduces a relatively large hyperparameter space, which can increase the tuning burden for deployment on new datasets, although this is partially mitigated by the model's generally low sensitivity around practical defaults.

---

> > ### Author Rebuttal · Reviewer_FEgi · 2026-04-04
> >
> > Thank the authors for their detailed rebuttal. They have addressed my technical concerns and I will maintain my score.

---

### Decision · Program_Chairs · 2026-04-30

**Decision:**

Accept (regular)

**Comment:**

This paper proposes an optimal transport–guided flow matching method (OPTION) for incomplete and unaligned multi-view clustering. By integrating conditional flow matching, Gromov–Wasserstein loss, and optional contrastive learning, the method provides a unified framework to handle missing views and cross-view misalignment, with a well-motivated and technically sound design.

All four reviewers provided positive evaluations, recognizing the novelty and technical contributions of the work, and their concerns were adequately addressed through the authors’ responses. Overall, there is a consistent consensus among the reviewers. Considering both the paper contributions and the strength of the rebuttal, the work meets the ICML standards in terms of novelty, technical rigor, and experimental validation. I recommend acceptance.